# The relative role of direct orbital forcing versus CO$_2$ and ice feedbacks on Quaternary climate

C. J. R. Williams [1,2] ✉, N. S. Lord[3], A. T. Kennedy-Asser [1], X. Ren[1], D. A. Richards [1], M. Crucifix [4], A. Kontula[5], M. Thorne [6], P. J. Valdes [1], G. L. Foster[7], R. M. Brown[7], E. L. McClymont [8] & D. J. Lunt [1]

During the Quaternary (the last 2.58 million years), Earth's climate has fluctuated between glacials and interglacials, paced by external forcings and mediated by internal feedbacks. However, General Circulation Models (GCMs), essential for addressing the mechanisms associated with these fluctuations, require substantial computational resources, meaning they are unsuitable for exploring orbital-scale variability on million-year timescales. Here, we use a GCM to calibrate a faster statistical model, or emulator, and apply this to the Quaternary. We show a good agreement between the emulated climate and proxy data over the last 800,000 years, especially the timing of glacial-interglacial cycles. A series of sensitivity experiments allows us to identify the dominant components driving long-term climate change. The results show that a combination of the CO$_2$ and ice sheet feedbacks provide the dominant contribution to the annual mean temperature signal, with the direct orbital radiative forcing playing only a minor role.

For the last few million years (Myr), the Earth's climate has fluctuated between cooler (glacial) and warmer (interglacial) intervals, as revealed by proxy records of palaeoclimate[1]. The timing and magnitude of these glacial-interglacial variations are forced solely by changes in the Earth's orbit and axis inclination around the sun, the three relevant parameters of which are precession, obliquity and eccentricity (which have main periods of ~ 23 thousand years (kyr), ~ 41 kyr, and ~ 96 and 400 kyr, respectively[2,3]. The direct orbital forcing is mediated by internal feedbacks in the climate system, including changes in the concentration of atmospheric CO$_2$ and variations in the extent and thickness of global ice sheets themselves. For the Quaternary (2.58 million years ago (Ma) to present day), comprising the Pleistocene, Holocene and, as some argue, the Anthropocene[4]), glacial-interglacial fluctuations are evident in a relatively large number of palaeoclimate records, constructed from various climate proxies[3]. However, although proxies for the Quaternary climate provide a clear

picture of climate variability, especially over the last 800 kyr, the interplay between forcings and feedbacks (associated with, for example, changes in the extent and thickness of ice sheets), is not well known[5]. Importantly, it is still not clear what proportion of the signals of Quaternary climate change is driven directly by the radiative effect of orbital forcing, and how much is associated with CO$_2$ and/or ice sheet feedbacks, which mediate and amplify that orbital forcing. Although this question of external forcings versus internal feedbacks has been investigated by previous studies, because of computational constraints (discussed below), almost all have only looked at certain periods during the Quaternary, such as the Last Glacial Maximum[6] (LGM, ~ 21 kyr ago (ka)), or have focused on shorter timespans, such as the last ~ 400 kyr[7].

To simulate changes in past climate that are physically-based and spatially resolved, climate models are required. More specifically, fully-coupled General Circulation Models (GCMs), or Earth System Models

[1]School of Geographical Sciences, University of Bristol, Bristol, UK. [2]Department of Geography, University College London, London, UK. [3]Fathom, Bristol, UK. [4]Earth and Life Institute,, Université Catholique de Louvain, Louvain-la-Neuve, Belgium. [5]Teollisuuden Voima Oyj, Eurajoki, Finland. [6]Mike Thorne and Associates Limited, Quarry Cottage, Hamsterley, Bishop Auckland, Co, Durham, UK. [7]School of Ocean and Earth Science, University of Southampton, Southampton, UK. [8]Department of Geography, Durham University, Durham, UK. ✉e-mail: c.j.r.williams@bristol.ac.uk

(ESMs) have proved useful for investigating the driving mechanisms, dynamics, feedbacks, and sensitivity of the climate system associated with variations in orbital forcing, $CO_2$, and ice sheets[8]. However, because of their structural complexity, along with their relatively high spatial and temporal resolutions, these models require substantial computational resources and time to run. As a result, they often cannot be run in fully-coupled transient mode over the long timescales (> 100 kyr) over which these forcings and feedbacks vary. Some studies have applied GCMs to conduct transient simulations of past climatic changes, but this has generally been in the form of a number of snapshot equilibrium simulations (the assumption being that the system is never far from equilibrium with the orbital forcings) in order to build a transient simulation[9]; however, even these simulations are typically limited to a single glacial-interglacial cycle. Alternatively, a limited number of fully transient GCM simulations have been conducted, such as in the framework of the Deglaciation PMIP experiment[10] or the PalMod project[11], but these in general simulate at most a single glacial-interglacial cycle. As such, simulations of multiple glacial-interglacial cycles with a full GCM or ESM are generally not practical, and the number of sensitivity studies that can be performed is limited. As far as we are aware, to date, only one GCM simulation can be considered fully transient[12], which involved running the Community Earth System Model version 1.2 (CESM1.2) for the entire Quaternary. This itself was an extension of earlier work[13], which ran the same model for the last 2 Myr to examine climate-driven habitat suitability of five hominin species. However, even this cannot be considered as a classic transient simulation, because the period was split into 42 chunks (of between 32–125 kyr each) that were run in parallel, and then concatenated afterwards to create a continuous timeseries[12]. In addition, the simulation also used the acceleration technique[14], in which some of the boundary conditions (such as the orbital parameters) are accelerated by a certain factor, resulting in a compressed simulation[12]; in this example, the acceleration technique resulted (using a factor of 5) in the 3 Myr simulation being compressed to 600,000 model years[12]. Arguably, a different (but complementary) approach is to use models with a lower complexity, such as Earth System Models of Intermediate Complexity (EMICs), to simulate long-term transient past climate changes[15]. However, EMICs provide a more schematic and/or lower spatial representation than GCMs, often with a relatively idealised representation of atmospheric processes that do not capture enough detail to understand climate variability, even at these long timescales.

Therefore, to resolve this competing desire for maximum process complexity and sufficient spatial resolution versus practical computing resource requirements and running time, statistical methods are being developed and applied to simulate long-term changes in climate. One emerging approach is that of emulators, which are statistical models that are calibrated on data from a more complex climate model, such as a physics-based GCM. They give a projection of the climate resulting from a certain set of input conditions (climate drivers), along with an estimation of the uncertainty (as shown by the variance at every timestep) associated with the projection. It should be noted, as a point of clarification, that while we refer to the emulator simulations in this paper, strictly data are interpolated rather than simulated. However, whilst the climate projection is based on the physics of the GCM and is at the same spatial resolution, the requirements of the emulator in terms of computational resources and time are a fraction of those required for the full GCM, once the emulator has been trained on existing GCM simulations. This makes them useful for investigating processes and climatic changes occurring over long time scales of hundreds of kyr or longer, thus allowing for comparison to proxies for climate[16] and model validation[17]. Emulators have been applied in sensitivity analyses of climate to orbital, atmospheric $CO_2$, and ice sheet configurations[18,19] and to investigate parametric uncertainty in models[20].

Here, we apply a climate emulator to simulate and understand the evolution of long-term climate change in the Quaternary. In short, the emulator is initially trained on a number of GCM simulations designed to capture the full range of climate conditions (including global sea level, GSL) from very cold (e.g., the LGM) to very warm (e.g., the Pliocene, 5.33–2.58 Ma), and is driven by five components, each relatively well understood global metrics: three orbital parameters (based on astronomical predictions of planetary motions), a proxy-derived $CO_2$ component, and a proxy-derived GSL component, used to represent ice sheets. Many other feedbacks/processes are represented in the emulator because they are included in the GCM itself (e.g., atmosphere and ocean dynamics (including the Atlantic Meridional Overturning Circulation, AMOC) and thermodynamics, clouds, sea ice, and vegetation). However, some feedbacks/processes are not represented in the GCM, and are therefore not included in the emulator, including dust, methane, lakes and freshwater fluxes associated with changes in ice sheet volume. Likewise, it should be noted that the emulator is a purely statistical model designed to replicate the output of the GCM. The emulator does not explicitly include any representation of any physical processes, but, as explained above, because they are represented in the GCM, the output of the emulator does implicitly take account of these processes. In principle, any variable that can be output by the GCM can be emulated, but here we choose to emulate near-surface air temperature (SAT) and mean annual precipitation (hereafter PREC), because these are the variables that can most robustly be evaluated with proxy data. A full description of the emulator, the process of creating the training data, and then building, optimising, testing and running the emulator, is described in the Methods. After a brief validation exercise in which the emulator output is compared to proxy data, the main purpose here is to use the emulator to answer the above question i.e., what proportion of Quaternary climate change is directly orbitally-forced versus $CO_2$ and/or ice sheet feedbacks, something which the efficiency of this approach allows given that the emulator takes several orders of magnitude less computational cost (see Methods, in particular Efficiency and computational cost) than if a climate model alone were used.

## Results

### All drivers simulation ($E_{11111}$): model-data comparison

Figure 1 compares the proxy and emulated temperature data over the last 2.58 Myr when all forcings and feedbacks are included in the emulator (simulation $E_{11111}$; see Methods, and in particular Section Sensitivity of emulated results to dominant driving input for more details, but in short this is when all forcings and feedbacks in the emulator are switched on, in comparison to e.g., $E_{00000}$ where they are all switched off), at one location in Antarctica from the Dome C ice core, providing δD-based annual-mean SAT, and four Ocean Drilling Programme (ODP) sites providing sea surface temperature (SST) from the $U^{K}_{37}$'-alkenone proxy (see Table 1 for details, Methods for the selection criteria used to obtain the sites, and Supplementary Fig. S1 in the Supplementary Material for the geographical location of the sites).

The results from Fig. 1 are discussed further below in quantitative terms but, qualitatively, the timing and duration of many of the interglacials and glacial maxima are similar in the model and proxy reconstructions, and the agreement between the magnitude of variations is generally very good (Fig. 1). These results (focusing on just the last 800 kyr) are also shown as a scatterplot in the Supplementary Material (Supplementary Fig. S2), where it can be seen that the relationship between emulated and proxy temperature varies according to location but is strongest at Dome C ($r^2 = 0.73$). At many of the locations there are instances when the relative maxima and minima are not well reproduced by the emulator, such as the interglacials at approximately 330 kyr, 240 kyr, and 125 kyr before present (BP), corresponding to Marine Isotope Stages (MIS) 9, 7 and 5, respectively (Fig. 1). Again, this is further demonstrated by Supplementary Fig. S2, where these

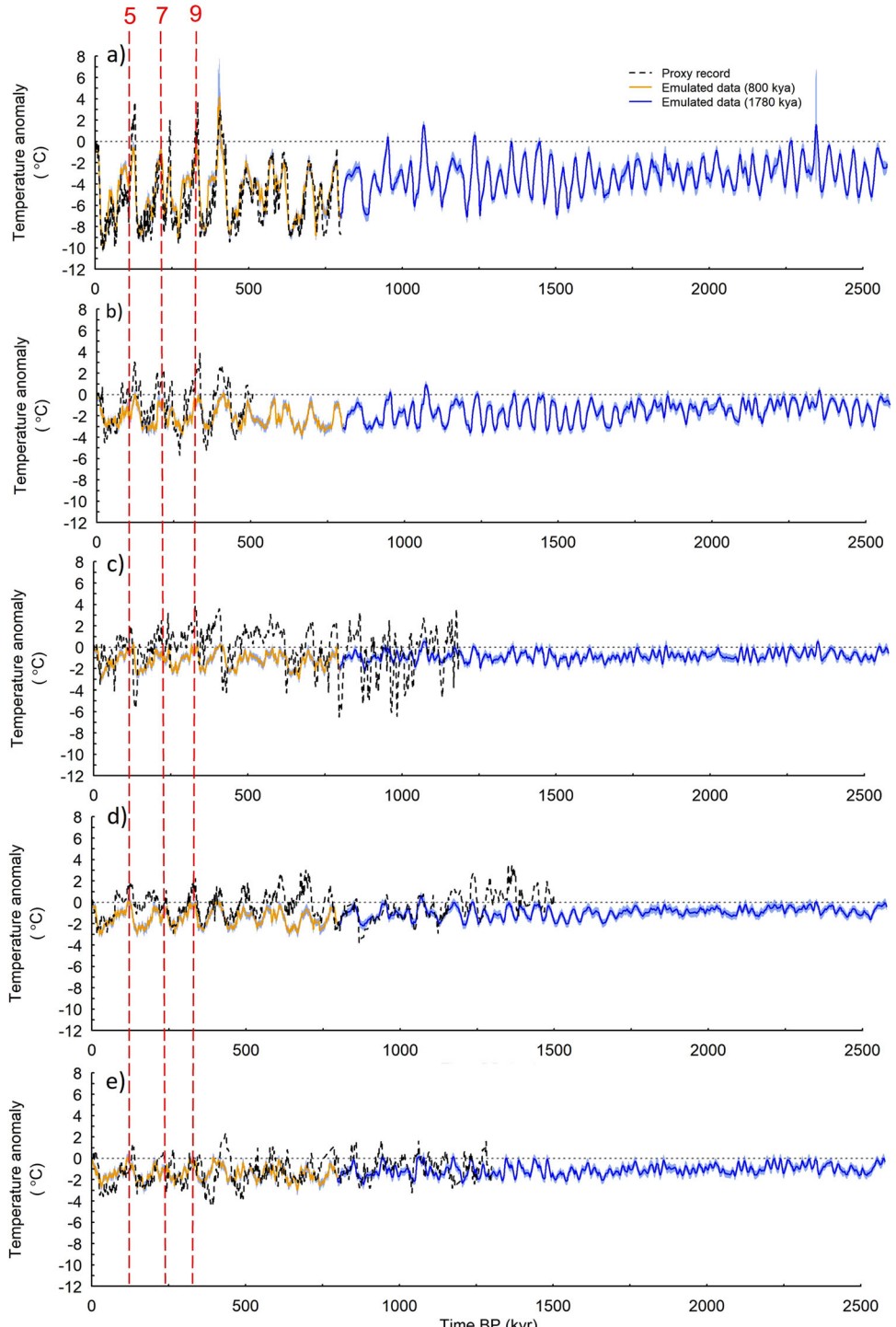

**Fig. 1 | Emulated versus proxy data comparison: temperature.** Timeseries of temperature anomalies (compared to a preindustrial control simulation (i.e., 0 thousand years (kyr)), °C) for the last 2.58 million years (Myr) at five locations, reconstructed from proxy data, where available (dashed black lines), and modelled every 1 kyr using the emulator $E_{11111}$ simulation (solid lines, either orange for the last 800 kyr or blue for the remaining 1780 kyr), with shading giving an estimate of uncertainty (see Discussion): (**a**) surface air temperature (SAT) from the Dome C ice core, Antarctica; (**b**) sea surface temperature (SST) at Ocean Drilling Programme (ODP) 1012, north-east Pacific; (**c**) SST at ODP 1123, central South Pacific; (**d**) SST at ODP 1239, Equatorial East Pacific; (**e**) SST at ODP 806, Equatorial West Pacific. See Table 1 for details. All SST and SAT shown as an anomaly compared to an average of the first 3 kyr from the dataset. Red vertical lines and numbers show Marine Isotope Stages (MIS).

interglacials appear warmer in the proxy data relative to the emulated temperature, matching the biased periods identified above in Fig. 1. At Dome C (Fig. 1a) there is a relatively large amount of SAT variability in the glacial-interglacial cycles, which continues back in time until approximately 1.5 Myr and then decreases even further back in time

towards the early Quaternary (in line with the other ODP sites, Fig. 1b–e); possible reasons for this are discussed below. At Dome C in particular (Fig. 1a), where the temporal range of proxy data is currently limited by existing drilling methods, the emulator gives an indication of what deeper (i.e., older) temperature reconstructions (as pushed for

**Table 1 | Sea surface temperature (SST) and surface air temperature (SAT) proxy data names, proxy type and Ocean Drilling Programme (ODP) locations (both actual locations, and then nearest grid box on the climate model grid), in descending order of arcsin Mielke (M) scores (between emulated and proxy data; see Methods and in particular Metric for quantifying model performance)**

| Name | Proxy/method type | Actual location | | Nearest location | | M scores |
|---|---|---|---|---|---|---|
| | | Lat (°N) | Lon (°E) | Lat (°N) | Lon (°E) | |
| Dome C[42,43] | $^2$H/$^1$H or δD (SAT) | − 75.1 | 123.3 | − 75 | 123.75 | 620.02 |
| ODP 1012[54] | $U^K_{37}$ (SST) | 39 | − 128 | 40 | − 127.5 | 327.96 |
| ODP 1239[55] | $U^K_{37}$ (SST) | − 0.672 | − 82.081 | 0 | − 82.5 | 200.48 |
| ODP 806[56] | MgCa (SST) | 0.318667 | 159.361 | 0 | 157.5 | 128.86 |
| ODP 1123[57] | Modern Analogue Technique (SST) | − 41.786 | − 171.499 | − 42.5 | − 172.5 | 128.21 |

by consortia such as Beyond EPICA, https://www.beyondepica.eu/en/) throughout the Quaternary might resemble there, if the $CO_2$ used to drive the emulator[21], is realistic.

The results presented here are in line with the previously-discussed simulation[12]. Concerning magnitudes of change (relative to the preindustrial, PI), they found a global-mean glacial cooling of 5.7 °C during the LGM in their simulation, consistent with their combined proxy-model estimate of 6.3–5.6 °C cooling[12]. This is similar, albeit slightly colder, to our results; here the emulator shows a global mean cooling of 5.1 °C during the LGM. This is discussed further in the Supplementary Material (Section S1 and Supplementary Fig. S3). In Antarctica, however, the results presented here are in better agreement with the previously-discussed simulation; an LGM cooling of 9.5 °C[12] compared to ~10 °C here (Fig. 1a). Concerning temporal trends, a clear decrease in global mean temperature anomalies and interannual variability in shown in the previous simulation[12] until the Mid-Pleistocene Transition (MPT, 1.25-0.7 Ma), after which the trend stabilises and variability increases[12]. Here, although we show the same change in variability before and after ~1 Myr, the emulated temperatures are not showing the long-term decline at any of the record locations (Fig. 1). Possible reasons for this are discussed below.

Figure 1 demonstrates that the proxy data show more glacial-interglacial amplitude than the emulator, especially in SST prior to 1 Myr (e.g., Fig. 1c–e). In fact, for the emulator to reproduce the variability shown by proxy data (at each individual location), a more extreme synthetic $CO_2$ curve is needed as input to the emulator. To obtain this (and again taking advantage of its efficiency), the emulator was run multiple times with idealised (constant) $CO_2$; the resulting temperature, in conjunction with the proxy data (using the same five locations as in Fig. 1), was used to generate timeseries of $CO_2$ via interpolation of the available data; for details see Methods (Sensitivity of emulated results to CO2), and Supplementary Fig. S4 in the Supplementary Material for the temperatures associated with the idealised $CO_2$. The resulting synthetic $CO_2$ curves are shown in Supplementary Fig. S5 of the Supplementary Material (for just the last 1600 kyr, corresponding to the maximum extent of the available proxy data), where the synthetic $CO_2$ can be seen against available reconstructed and modelled $CO_2$, at each of the five locations on which the synthetic $CO_2$ is based. In short, a comparison between the synthetic $CO_2$ and actual $CO_2$ suggests that the emulator would need input $CO_2$ that, in some locations, is more than 1.5 times larger than its current driving $CO_2$ to match the variability in temperature shown by the proxy data (Supplementary Fig. S5). At Dome C, the synthetic $CO_2$ corresponds well in terms of variability to the observations, but the magnitudes are often greater than the proxy data, especially during the interglacials discussed above (Supplementary Fig. S5a). Elsewhere, in some locations (such as in the central South Pacific) the synthetic $CO_2$ is much greater in both magnitude and variability relative to either reconstructed or modelled $CO_2$ (Supplementary Fig. S5c). The observation that this synthetic $CO_2$ is very different in places to the driving $CO_2$ might be explained by: (i) the low variability shown by the emulator is correct

but the proxy temperature data are too variable; (ii) the variability of the proxy temperature data is correct but the $CO_2$ records driving the emulator are causing the resulting temperature to underestimate the correct amount of variability (this can only apply prior to the ice core record, because the ice-core $CO_2$ is assumed to be accurate); or (iii) using other additional driving components (such as those discussed above that are not included in the GCM) might improve this match.

Concerning PREC, the composite Chinese speleothem δ$^{18}$O data records (we acknowledge that this is not the same as PREC, but is used here as a first-order comparison for precipitation) are compared with emulated precipitation in Fig. 2. As with temperature, the emulator reproduces the major variations evident in the δ$^{18}$O record reasonably well, particularly with regards to the precession-driven timing of maxima and minima in precipitation. Unlike SAT and SST, however (where the level of variability decreases further back in time), a similar level of variability continues when the emulator is run further back beyond the proxy data records, and this is true at the three sites considered (Fig. 2).

To quantify the model-proxy comparison, we calculate the arcsin Mielke measure, or M score[22] to assess the goodness of fit between two variables with the same units, and therefore the skill of the emulator. The resulting skill score has a maximum possible value of 1000, while a score of 0 or negative values indicates no skill[22]. See Methods, in particular Metric for quantifying model performance. We calculate an M score between the emulated temperature and proxy data, for the all drivers simulation (in other words, a simulation in which all five drivers - three orbital parameters, $CO_2$ and GSL - varied according to proxy data, denoted $E_{11111}$) throughout the 2.58 Myr; see Table 1. SAT over Dome C is shown to be the best match with the proxy data, with an M score of 620 (out of a possible 1000, which represents an exact agreement). For SST at ODP locations, M scores are 328, 200, 129 and 128 (Table 1).

It is important to put these M scores into context. A point of reference is to calculate the M score for a constant climate signal with the same mean as the proxy record (i.e., keeping the driving $CO_2$, orbit, and ice as constant, averaged over the last 800 kyr; see Methods and in particular Section Sensitivity of emulated results at proxy locations for details). As expected, at all sites the modelled variations perform substantially better than a constant signal; for example, at Dome C the original emulated results gave an M score of 620 whereas the constant climate signal based on the mean gave an M score of 0.65, the constant climate signal based on the mean LGM gave an M score of 0.48, and a random climate signal gave an M score of −0.04. As a further contextualisation, we calculate an M score between the model and proxy at every possible combination of sites (see Supplementary Material and Supplementary Table S1). We assess agreement as good for a particular site if the best M score is obtained when the model results at that site are compared with proxies from the same site (i.e., the diagonals in Supplementary Table S1). Although some sites (such as Dome C and ODP 1239) show good agreement, other sites do not. However, this can be explained by the relative geographical proximity of some of

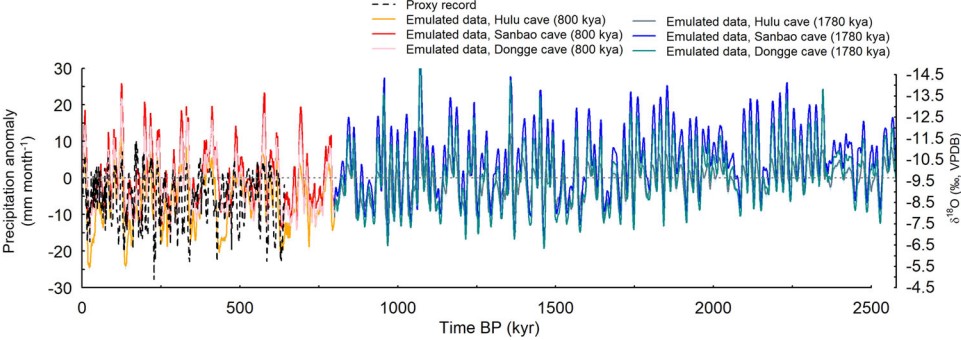

**Fig. 2 | Emulated versus proxy data comparison: precipitation.** Timeseries of mean annual precipitation (PREC) anomalies (mm month$^{-1}$) for the last 2.58 million years (Myr) in China, reconstructed from proxy data (dashed black line) and modelled every 1 thousand years (kyr) using the emulator $E_{11111}$ simulation (solid lines, coloured separately for the last 800 kya and the remaining 1780 kya). Proxy data are from a composite $\delta^{18}O$ record (‰) covering the last 640 kyr, constructed from records from the Hulu, Sanbao, and Dongge caves, where cave speleothem $\delta^{18}O$ data are taken to be a proxy for variations in the strength of the East Asian monsoon. To enable comparison, the Hulu and Dongge proxy records are plotted 1.6‰ more negative to account for their higher values (Cheng et al. 2016, Wang et al. 2001). Emulated anomalies are shown at the grid boxes containing the Hulu (grey/orange solid line), Sanbao (blue/red solid line), and Dongge (turquoise/pink solid line) caves.

the sites; proxy data at ODP 1012, for example, are giving the best match to emulated data at ODP 1239 (M score = 360), both of which are in the eastern Pacific. Likewise, agreement is sometimes best when the sites are geographically distant but, for example, at a similar latitude.

## Single driver experiments

Following the model-data comparison, the relative impacts of the individual drivers on emulated SAT were investigated. To overcome potential disadvantages of linear factorisation (see Section S2 in the Supplementary Material text, and Supplementary Figs. S6, S7, for the results), a linear-sum/shared-interaction factorisation was also conducted[23], again globally. This is shown in Fig. 3, with the results being very similar (but more robust) compared with the linear factorisation (Supplementary Fig. S7).

The key finding is that the direct impact of the orbital parameters (i.e., obliquity, eccentricity and precession) on the M scores is negligible and they are thus contributing little to the overall signal (Fig. 3b–d). Of the three orbital parameters, obliquity provides the highest contribution to the overall signal, particularly over high latitudes (e.g., the Southern Ocean) with M scores reaching ~200–300 as opposed to ~100 or less from eccentricity and precession (Fig. 3b–d). It should be noted that eccentricity and precession are combined as two forcings, one of ecosω and one of esinω, consistent with previous work[18] and motivated by the fact that eccentricity modulates the precession forcing. Also, as discussed in more detail below (see Discussion), we are not suggesting that the overall orbital forcing is negligible, but that the direct orbital radiative forcing on the annual mean temperature response is negligible. The orbital forcing does have a very significant indirect impact, through controlling the $CO_2$ and ice sheet forcings (which, given that these are used to drive the emulator, therefore account for the indirect impact of orbital forcing), and also has a significant direct impact on seasonal precipitation (e.g., the West African monsoon). In contrast to this negligible orbital contribution, the results also show that $CO_2$ and ice are contributing the most to the overall signal, particularly over the tropical landmasses (for $CO_2$, Fig. 3a) and polar regions (for ice, Fig. 3e). When averaged globally $CO_2$ is contributing 54% to the overall signal and ice is contributing 35%, whereas the orbital parameters' contributions are all < 5% (Fig. 3f).

## Discussion

Generally, there is good agreement between emulated and proxy-reconstructed temperature and precipitation, and the temperature results presented here are in good agreement with the previously-

discussed simulation[12]. Magnitudes of change are similar, especially for temperature at Dome C, and a general decrease in temperature variability prior to the MPT is shown in both studies. There are some discrepancies between emulated and proxy data, however, particularly towards the beginning of the Quaternary[12]. Their simulation also diverges considerably from their comparative SST proxy data prior to the MPT, not only in terms of representing the orbital phase but also failing to capture the long-term extratropical Pleistocene trends. The same result is shown here, with a considerably lower amplitude variability from the emulator during this early period, relative to the last 800 kyr. In contrast to the previous simulation[12], however, the gradual decrease in global temperature change during the beginning of the Quaternary until ~1 Ma, as shown by their simulation, is not reproduced here. Although the results presented here show a decrease in variability during the early Quaternary, no long-term trend is obvious. However, this can be explained by the driving inputs to the emulator, where our GSL and $CO_2$ are also much less variable than in the previous simulation[12] during this early period (see Supplementary Material and Supplementary Fig. S8). A long-term lowering of sea level (implying cooling) is suggested by the driving GSL parameter used here (Supplementary Fig. S8), but it is not as evident as the temperature trends[12] and, importantly, is not shown in the driving $CO_2$ parameter used here. Given that, as shown by the sensitivity simulations (discussed further below), $CO_2$ is the dominant driving forcing and the impact of the GSL parameter is limited primarily to polar regions, this possibly explains why we are not seeing a long-term decline in global mean temperatures, i.e., there is no obvious long-term change in $CO_2$, the dominant driver of the emulator.

This signal, or lack thereof, in the driving input parameters (or forcings) also explains why the emulator shows no obvious step in the timeseries coinciding with the MPT. Given the fundamental change in glacial cycles that occurred during the MPT, one might expect to see a corresponding change in temperatures. Although, as discussed above, emulated temperature variability is clearly lower during the early Quaternary relative to the last 800 kyr, no obvious change during the MPT is shown by the emulator. This is because, as above, there is also no clearly defined transition during the MPT in any of the driving input parameters, especially the dominant $CO_2$ forcing. A change in variability is shown, and is therefore reflected in the emulated temperatures, but no clear transition occurs here during the MPT.

The relatively large uncertainty in the emulator (shown by the blue shading in Fig. 1) at many of the sites during the most recent interglacials, such as at Dome C (Fig. 1a), is likely due to, contrary to the transition from interglacial to glacial conditions for which changes in

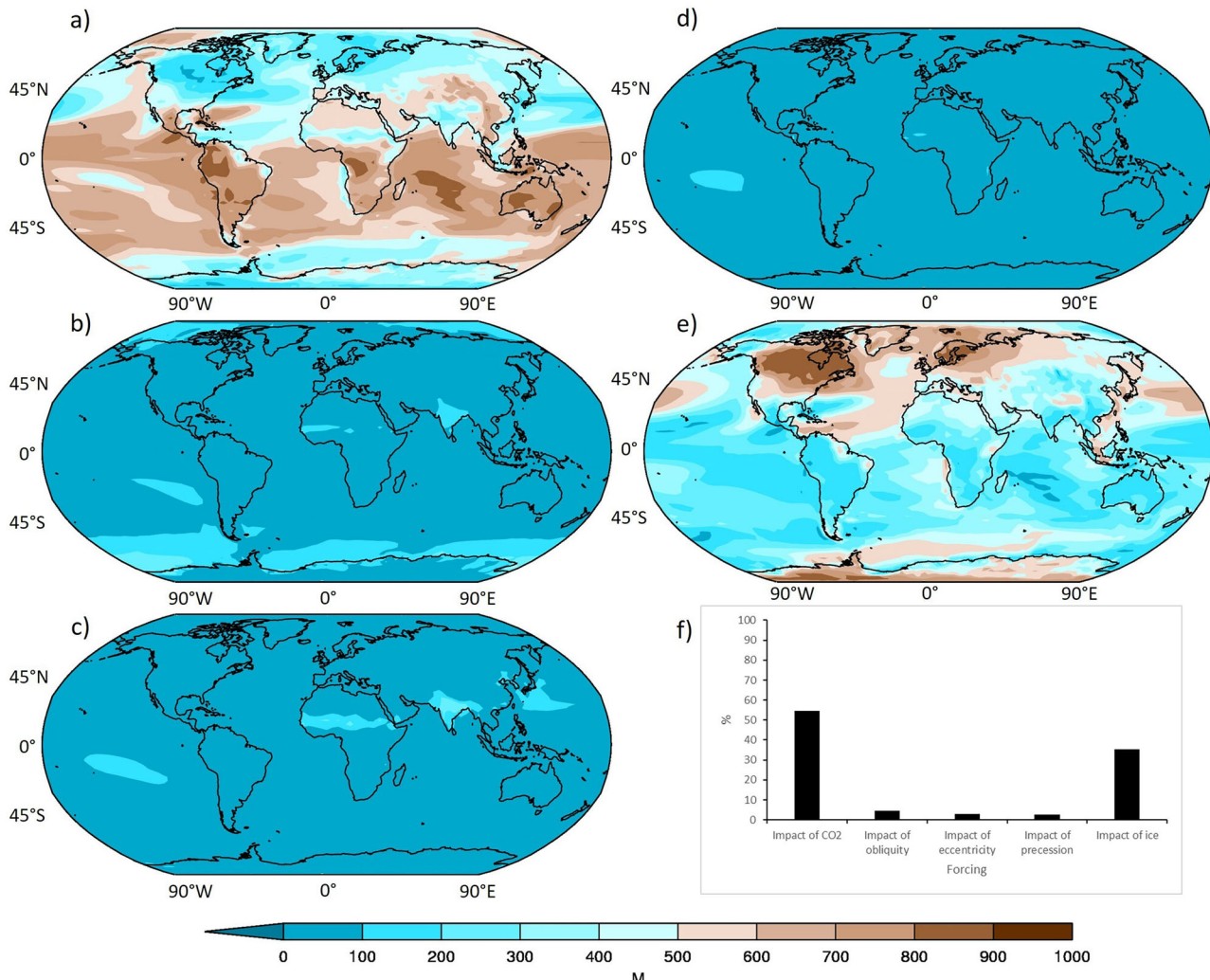

**Fig. 3 | Maps of linear factorisation.** Arcsin Mielke (M) scores between emulated temperature from the $E_{11111}$ simulation and that from individual simulations following the linear-sum/shared interaction factorisation of Lunt et al. (2021), averaging over all 120 pathways for each driving component: (**a**) atmospheric common dioxide ($CO_2$); (**b**) Obliquity; (**c**) Eccentricity; (**d**) Precession; (**e**) Ice; (**f**) Global means, expressed as a percentage. M scores, in short, are the quantitative metric used here to assess the goodness of fit, in this case between the simulation with all forcings included and those from each of the driving components. M scores represent the mean square error non-dimensionalised by the variance; a score of 1000 is a perfect match, whereas a score of 0 or negative values indicates no skill. Note that the sum of (**a**–**e**) is by definition equal to 1000 (See Methods for more details).

the ice sheets are well sampled, there are only two different Antarctic and Greenland ice sheet configurations for calibrating interglacial conditions; modern-day ice (modice) and reduced ice (lowice). To clarify, this uncertainty is purely related to how well the emulator can be expected to reproduce an equivalent result from the GCM that was used to train it. The emulator uncertainty in regions where the ice sheets change significantly (whilst in an interglacial state) is relatively high because the emulator effectively only has two ice sheet configurations to interpolate between. At MIS 11, GSL is relatively high for an interglacial, suggesting increased melting of one or more of the polar ice sheets, whereas most of the other interglacials have GSL values that are lower than the PI, and therefore this is less of an issue. Concerning precipitation, whilst there do appear to be similarities between the emulated precipitation and observed $\delta^{18}O$, direct quantitative comparison between these records is challenging without including an explicit representation of oxygen isotopes in the underlying GCM simulations.

The experiments presented here can also be compared to previous work isolating contributions from individual forcings, such as snapshot simulations[24] or transient simulations[25]. But perhaps the most appropriate comparison is with work which carried out idealised

simulations using the Geophysical Fluid Dynamics Laboratory (GFDL) Climate Model 2.1 (CM2.1) fully-coupled GCM[8], using the same drivers and a comparable methodology to that applied here. This found a reasonable fit between the simulated and reconstructed data[8], consistent with the results presented here, however in contrast to us this found the best match at their locations in the eastern and western equatorial Pacific[8]. The finding of an underestimation (overestimation) of temperature variability in the eastern (western) Pacific[8] is roughly consistent with our results here. At Vostok, relatively close to Dome C presented here, they observe substantial differences between the simulated data and proxy record, which was attributed to obliquity variations and the model not being able to correctly capture the obliquity response[8]. In contrast, the results presented here suggest relatively good agreement at this location, and indeed the best agreement among all the proxy data locations (e.g., Fig. 1a), possibly suggesting that our emulator is more successfully capturing these orbital signals.

Secondly, the same study conducted a similar factorisation analysis, suggesting that the ice sheets drive greater variation than $CO_2$, particularly during glacial conditions[8]. It did, however, come to a conclusion similar to ours concerning the direct orbital forcing

accounting for the least amount of temperature variability[8]. This has also been suggested by other work, such as that focusing on the LGM; a negligible global mean cooling is suggested when switching to LGM orbital forcings only, with most of the cooling coming from ice sheet and greenhouse gas feedbacks[26]. Given that the direct orbital forcing contributes so little to the overall temperature signal, it could be argued that constraining orbital forcing (as is done by many model intercomparison studies, such as the Paleoclimate Modelling Intercomparison Project (PMIP)) is unnecessary. However, it is important to note that here we are focusing on annual mean temperatures. If other, more seasonally-dependent processes are investigated, such as the work which discussed the effect of orbital forcings and $CO_2$ on Southern Hemisphere climate (and in particular its westerlies)[7], then orbital forcings become much more directly related. Likewise, if the aim was to examine, for example, monsoon precipitation variability, a summer and winter version of the emulator would need to be built (separately), in which case the orbital component should have more of an impact. We actually do see a much stronger, orbit-only signal in the emulated precipitation, but this was not included in the various factorisation approaches (which were based on annual-mean temperature only).

There are a number of limitations associated with the methodology presented here. Firstly, discrepancies between the emulator and proxy data could occur due to errors in: (i) the emulator, where the response of a climate variable to a driver has not been fully captured; (ii) errors in the underlying GCM, which may propagate into the emulator, such as changes in atmospheric dust (which, for example, may have an impact on long-term climate[27] but are associated with significant uncertainty and are not accounted for in the GCM); (iii) missing processes in the emulator, such as freshwater fluxes (meaning millennial-scale climate change driven by fresh water forcing, such as Dansgaard-Oeschger cycles[28] or Heinrich events[29], are not directly simulated, although their impact on $CO_2$ and ice sheets is accounted for in this framework in which $CO_2$ and ice are prescribed from observations) or non-$CO_2$ greenhouse gases and aerosols; and iv) uncertainties in the proxy data, such as their calibration or age models[30] or seasonal biases[31]. Secondly, the GCM simulations on which the emulator is calibrated use the ICE-5G ice sheet reconstruction[32]. However, other reconstructions of the ice sheets also exist[33] and demonstrate some differences to the ICE-5G reconstructions (see Section S3 in the Supplementary Material for more details). Thirdly, the issue of hysteresis is not considered here, because we are judging each climate state independently; we acknowledge, however, that this is an important consideration (as has been demonstrated by many other studies investigating individual processes[34]) and could be included in future work. Lastly, the emulator is based on the results of a single GCM, HadCM3, whereas other models may produce different results, due to variations in model structure and the parameterisations used.

In conclusion, SAT and PREC is presented here for the entire Quaternary (2.58 Myr), produced using a statistical emulator calibrated on a large ensemble of HadCM3 simulations, and driven by direct orbital forcing, and atmospheric $CO_2$, and ice volume feedbacks. Following a model-validation exercise, in which the emulator results at various locations were compared to a number of proxy records for palaeoclimate taken from those same locations, the emulator was used to test the sensitivity of climate to the individual forcings.

Several conclusions can be made. Firstly, the model-data comparison results suggest a good level of agreement between the reconstructed and emulated temperature and precipitation, particularly during the last 800 kyr when the driving variables (especially $CO_2$) are most reliable and particularly in terms of the timing and duration of many of the interglacial and glacial periods. The timings and relative minima and maxima in precipitation are also well reproduced by the emulator in general. There are some discrepancies for both temperature and precipitation, however, such as instances when relative minima and maxima are not in agreement with the proxy data; these are likely to be the result of a combination of errors in the emulator, the forcing drivers, the underlying GCM, and the proxy data. Secondly, considering the entire Quaternary, the emulator performs reasonably well compared to available proxy data, despite greater uncertainty the further back in time from both sources. The emulator has at least some regional skill in terms of magnitude and cyclicity of climate change, and therefore, it is possible that the emulator is able to give a projection of what certain proxy data, such as at Dome C in Antarctica, might resemble if deeper drilling occurs. Coupled with the general agreement (where proxy records exist) as the emulator is run further back in time, the results indicate that it is appropriate to use the emulator to explore the relative importance of the various drivers that contribute to Quaternary change.

Returning to the main research question of the dominant drivers of Quaternary climate change, the factorisation simulations show that the driving components contributing the most to the overall signal are $CO_2$ and ice, in that order, with the direct orbital forcing providing a much lower contribution. Regionally, the $CO_2$ signal is primarily dominant over the tropical regions and, in particular, tropical land-masses such as South America and southern Africa, whereas the ice signal is primarily dominant over high latitudes and polar regions, especially Antarctica and Greenland. Although the all drivers simulation does not provide the best match with proxy data at all locations, the simulations that do provide the best match all have varying $CO_2$ and ice sheets in common. The only place where the all drivers simulation does provide the best match with proxy data is at Dome C (which also gives the highest M scores compared to the other locations), and this is likely because of the dominant ice driving component, which would have the clearest signal over this region. Although the three orbital components, either individually or combined, do provide some contribution to the overall signal (4, 3 and 3% for obliquity, eccentricity and precession, respectively), they are overshadowed by contributions from $CO_2$ and ice (54% and 35%, respectively).

## Methods
An emulator is a statistical representation of a more complex model, in this case, a GCM. The underlying principle is that a relatively small number of simulations are carried out initially using the GCM, which sample the multidimensional climate driver input space (using Latin hypercube sampling, see below); in this case, five dimensions consisting of three direct orbital forcing dimensions, a $CO_2$ feedback dimension, and a GSL feedback dimension. The emulator is calibrated on (or tuned/trained to) these GCM simulations, including the full two-dimensional latitude-longitude global coverage of the grid GCM data, with the aim of being able to interpolate the GCM results such that a prediction is made of the output that the GCM would produce if it were run using any particular combination of input climate drivers[17]. The emulator also produces an estimation of the uncertainty associated with the prediction. For any single combination of values of these climate forcings, the emulator simulates the corresponding state of a particular climate variable, in this case, global two-dimensional fields of the SAT and PREC. As such, temperature and precipitation emerge as output from the emulator for any combination of input forcings. The emulator can be validated by comparing results with additional GCM results not included in the original training stage[17]. The combination of the emulator and the GCM itself can be validated by comparing the resulting output to existing proxy reconstructions (i.e., a model-proxy validation). The process of building, optimising, testing and running the emulator is described below, but first, a description of the emulator's efficiency and low computational cost is provided.

## Efficiency and computational cost

To run over the full 2.58 Myr at a temporal resolution of 1 kyr, calculate comparisons with the proxy data, and produce output files and figures, the emulator ran at several orders of magnitude faster than a standard GCM (minutes, as opposed to months or years). As a comparison, for a GCM such as HadCM3 to carry out the same process (i.e., produce a quasi-transient simulation of 2,580,000 years), even at its optimal speed of approximately 120 model years/day, it would take ~ 59 years to run. Even the shorter, 800,000-year simulation would take just over 18 years to run. Therefore, carrying out a single quasi-transient simulation of the entire Quaternary using this model, let alone the 31 additional sensitivity simulations of the late Pleistocene, would be impossible. It should be noted that prior to running the emulator, 182 HadCM3 simulations were undertaken to provide the training data for the emulator. The computationally expensive part of the process, therefore, is the creation of the training data (i.e., the HadCM3 simulations), but this only needs to be done once, at the beginning; thereafter, for any quasi-transient simulation or sensitivity tests, the emulator is several orders of magnitude faster than a GCM. Moreover, the emulator can be run, at this speed, on a standard desktop PC, whereas HadCM3 requires resource-intensive super-computing infrastructure, clearly demonstrating the efficacy of the emulator and its suitability for subsequent work.

**Step 1: Creation of training data.** The emulator was calibrated on simulations run using the HadCM3 climate model, a fully coupled atmosphere-ocean GCM developed by the UK Met Office. The specific model setup of the GCM is denoted HadCM3B-M2.1aE[35]; hereafter referred to as HadCM3. The horizontal resolution of the atmospheric component is 2.5° latitude by 3.75° longitude with 19 vertical levels, whilst the ocean has a resolution of 1.25° by 1.25° and 20 vertical levels. Please see Supplementary Material (Section S4) for more details on HadCM3.

Overall, a 182-member ensemble of HadCM3 simulations was used to train the emulator, with input parameters varying across five dimensions: three main orbital parameters of longitude of perihelion ($\varpi$), obliquity ($\varepsilon$) and eccentricity (e), atmospheric $CO_2$ concentration (from ~ 180 ppmv to ~ 1900 ppmv) and GSL (from − 100 m to + 30 m relative to modern). This is one more dimension than a previous study based on these simulations, which did not include GSL[17]. The 182-member ensemble consisted of two 60-member sub-ensembles (modice and lowice) and a 62-member sub-ensemble (highice).

The two 60-member ensembles had ranges of orbital and $CO_2$ values (see Supplementary Table S3) that were appropriate for a wide range of palaeo and future climate states[17]. Eccentricity and longitude of perihelion were combined under the forms $e\sin\varpi$ and $e\cos\varpi$, given that, in general, at any point in the year, insolation can be approximated as a linear combination of these two terms and obliquity[36]. Latin hypercube sampling was used to optimally fill the 4-dimensional input space of orbit and $CO_2$. The two 60-member sub-ensembles differed solely by having two different fixed ice sheet configurations, consisting of changes to the Greenland and West Antarctic ice sheets (GrIS and WAIS, respectively) only. One sub-ensemble had present-day ice sheet extents, denoted the modice sub-ensemble, and the other had reduced extents, based on the PRISM4 mid-Piacenzian (a warm interval, ~ 3.3-3.0 Ma, during the Pliocene) reconstructions[37], denoted the lowice sub-ensemble. The lowice sub-ensemble was created from the modice sub-ensemble using the anomaly approach[17]. A third sub-ensemble consisting of 62 GCM simulations was also run with HadCM3; a series of snapshot simulations covering the last glacial cycle (LGC, ~ 115-11 kyr before present (BP)), forced by appropriate changes in orbit, atmospheric $CO_2$ concentration and ice sheet evolution, denoted the highice sub-ensemble[24]. All of the HadCM3 simulations were initialised from preindustrial climate conditions and run for several hundred years, by the end of which they had all reached a quasi-equilibrium state at the surface and subsurface; the training data used here, therefore, can be considered independent of initial conditions, as each ensemble member represents a stable climate state.

## Step 2: Building the emulators

**Creating the interglacial and glacial training data.** The emulator described here is an updated version of a previous emulator[17], updated firstly by including an additional driving term of ice sheet volume (using GSL as a proxy for this), and secondly by adding additional HadCM3 simulations to the calibration that included various ice sheet configurations (sub-ensemble highice, as described in Step 1 above). In practice, two separate emulators were used, one calibrated on the lowice and modice 60-member sub-ensembles and used for interglacial conditions (defined as GSL > = 0 m), and one calibrated on the 60-member modice and 62-member highice sub-ensembles and used for glacial conditions (defined as GSL < 0 m). Taken together, therefore, the HadCM3 simulations cover the full range of GSL, including LGM, glacial-interglacial cycles and Pliocene-like conditions. Please see Supplementary Material (Section S3) for more details on creating the training data.

The approach of having two separate emulators (shown in the Supplementary Material, Fig. S9) with separate training data for the interglacial emulator applied for GSL > = 0 m compared with the glacial emulator for GSL < 0 m, was taken to ensure that there was no leakage of state-specific climate variations across the glacial-interglacial transition. The concept and effect of this leakage can be illustrated (see Supplementary Fig. S10 in the Supplementary Material) by considering surface air temperature for two climate states, one with an emulated climate with GSL = +5 m (Supplementary Fig. S10a, b), and one with an emulated climate with sea level = − 5 m (Supplementary Fig. S10c, d), firstly as produced by the separate emulators (Supplementary Fig. S10a, c, as used in this paper) and secondly produced by a combined emulator, in which all training data are used to train a single emulator (Supplementary Fig. S10b, d). For the + 5 m climate state, both the separate and combined emulators show clear warming associated with melt of the Greenland and Antarctic ice sheets (Supplementary Fig. S10a, b). This is correct, and appears because the lowice training data include ice sheet reduction in these regions, associated with the PRISM4 mid-Pliocene ice sheet reconstruction. However, for the −5 m climate state, there are clear differences between the results of the separate and combined emulator versions in these regions (Supplementary Fig. S10c, d). For the combined emulator (Supplementary Fig. S10d), the algorithm is still influenced by the Pliocene lowice training data, producing anomalous cooling in the regions that show melt in the lowice training data. However, the two separate emulators are, for this GSL, only influenced by the highice and modice training data, and so do not have this unphysical leakage across the glacial-interglacial transition (Supplementary Fig. S10c). In essence, the use of separate emulators is required because in the real climate system, warming relative to modern results in ice sheet loss primarily in the Greenland/Antarctica region, whereas cooling relative to modern results in ice sheet growth primarily in the Laurentide/Fennoscandian region. The use of two separate emulators, therefore, ensures that the different states are characterised separately and correctly. In addition, cheques were carried out to ensure that there were no discontinuities across the glacial-interglacial transition between the climate projections from the two different emulators. This lack of discontinuity is unsurprising, given that the modice ensemble was common to both emulators, and sits at the surface GSL = 0, on the boundary of the two emulators. We therefore consider it appropriate to use the two separate emulators, as combining them into one leads to higher uncertainties at the boundary between glacials and interglacials, due to the leakage effect discussed above.

The reconstructed ice sheet extents in the high ice ensemble are based on the ICE−5G model[32]. These reconstructions were primarily selected because the associated HadCM3 model simulations already existed[24]. The reconstructions are based on palaeo data (GSL, local sea level, and ice-sheet extent) and include the range of associated data that was required for this work. In addition, for glaciations prior to the last glacial cycle, there is very little or no palaeo data available that enables the global 3-D reconstructions which are required here. There are, however, other reconstructions (see Section S3 in the Supplementary Material); although we acknowledge the existence of these, the lowice, modice and highice ensembles used here, when combined, do capture changes in climate and ice sheet extent ranging from interglacial states to full glacial conditions

### Input parameter details and optimisation

In this study, a principal component analysis (PCA) Gaussian process (GP) emulator is used[38], with the subsequent Bayesian treatment[39,40] and a PCA approach[41]. Climate data for the entire global grid, from the aforementioned HadCM3 simulations, are used to calibrate the emulators, as opposed to calibrating separate emulators based on data for individual proxy locations. This approach is taken because, for the past climate, the global response overall is of interest, rather than just the response at specific locations individually. It also means that the results are consistent across all locations.

The driving parameters for the interglacial emulator can be expressed as a $120 \times 5$ matrix (n × p), representing $n = 120$ HadCM3 simulations and $p = 5$ input forcing parameters, as discussed above. These $120 \times 5$ input driving parameters are single values; the emulator is trained on the corresponding 120 spatial patterns of temperature and precipitation from the 120 simulations coming out of HadCM3, expressed as PCAs (see below).

The matrix containing the training data from HadCM3 had dimensions of longitude × latitude × $n = 96 \times 73 \times N$, for the interglacial ($N = 120$) and glacial ($N = 122$) emulator, respectively. A PC analysis was performed on each of the n GCM simulations, the results of which were then used to calibrate the two emulators. Five correlation length hyperparameters ($\delta$) were used, one for each of the five input factors, which describe the smoothness of the climate response in the HadCM3 data to the input conditions[17]. A nugget term ($\upsilon$) was also used, which accounts for any non-linearity in the output response to the inputs, non-explicitly specified inactive inputs and the effects of lower-order PCs that are excluded from the emulator. The optimal values for these hyperparameters and the number of PCs retained were calculated during calibration and evaluation of the emulator.

Four emulators were created in total, two of which were trained on SAT data and two of which were trained on PREC data, for an interglacial (trained on modice and lowice ensembles) and glacial (trained on modice and highice) version, respectively. Before being used, the emulator configurations were optimised, which involved generating an ensemble of emulators with different numbers of PCs retained (the majority of the PCs were discarded as they only account for a very small amount of total variation in the HadCM3 data) and different values for a number of hyperparameters used in the emulator, including the correlation length hyperparameters ($\delta$) and the nugget parameter ($\upsilon$). The performances of the different emulators were compared in order to identify the optimal number of PCs to retain and the hyperparameter values to use. It has been demonstrated that the optimisation can be carried out on SAT, and then the optimised configuration applied to other climate variables (e.g., precipitation), with no significant loss of performance[17]. Further details on the optimisation process are discussed in Section S5 of the Supplementary Material.

### Step 3: Testing the optimised emulator

**Leave-one-out approach.** The performance of the optimised emulators was evaluated, firstly by using a leave-one-out cross-validation approach. Here, a series of emulators was constructed and used to predict one left out HadCM3 simulation each time. For example, for the interglacial emulator (normally based on 120 HadCM3 simulations), the emulator was built on 119 simulations and used to reproduce the 120th, with the results then compared to the actual 120th HadCM3 simulation. This was then repeated for every possible combination ($N = 120$) of left-out simulations. The number of grid boxes for each experiment, calculated to lie within different SD bands, and the root mean squared error (RMSE) averaged across all the emulators, were used as a performance indicator. The results suggest that that the emulators are able to reproduce the left out ensemble simulations reasonably well (see Supplementary Fig. S11 in the Supplementary Material for individual measures of this e.g., the percentage of grid boxes for which the emulator predicts the SAT of the left out experiment to within 1, 2, 3 and > 3 SD), with no obvious systematic errors in their predictions.

### Last glacial maximum approach

One of the input training simulations in the highice ensemble is an LGM simulation. Because extensive proxy data exist for the LGM, this allows us to evaluate the HadCM3 climate model itself for this time period. At the same time, we can evaluate how well the emulator reproduces the HadCM3 LGM simulation (a less stringent test than the leave-one-out test described above, as the target simulation is part of the training data). This evaluation is discussed in Section S1 of the Supplementary Material.

### Step 4: Running the emulators

**Model-data comparison.** In addition to evaluating the emulator/ HadCM3 output with the aforementioned LGM reconstructions, another way of testing the emulator is to compare the output with timeseries of existing palaeo proxy data.

For SAT and PREC separately, the emulator was therefore run for the last 800 kyr (i.e., the late Pleistocene and Holocene). This was achieved by running the emulator 801 times, once for each 1 kyr timeslice between the present (i.e., 0 kyr) and 800 ka, each with appropriate values for the 5 driving parameters. For timeslices that had a GSL value of equal to or higher than 0 m (equivalent to present-day interglacial conditions), the interglacial emulator was used, whilst for timeslices that had a GSL value lower than 0 m, the glacial emulator was used. Hereafter, and throughout the main manuscript, these 4 emulators (two SAT, two PREC; two interglacial, two glacial) are known collectively as the emulator. For each timeslice, the input to the emulator is the five input forcing parameters, either calculated from astronomical solutions[2] (for the 3 orbital parameters) or from proxy data (for $CO_2$ and GSL), as described below. The resulting emulated latitude-longitude SAT and PREC were then compared with the proxy data at the location of the various sites, as described in the Results section. The last 800 kyr were initially selected because a range of high-resolution palaeo records exist, of multiple variables and in multiple locations, providing both forcing data for the emulator and proxy climate reconstructions which can be compared to the emulator results. However, some of the sites provide reconstructions going further back in time and so, once the above model-data comparison had been undertaken, an additional emulator simulation of 2.58 Myr (i.e., the entire Quaternary) was conducted.

### Forcing parameters used when running the emulator

The palaeo data used to force the emulator are illustrated in Supplementary Fig. S8 of the Supplementary Material. For the first 800 kya, orbital data (Supplementary Fig. S8a) were calculated at 1 kyr resolution[2]. For $CO_2$, a composite record of observed atmospheric $CO_2$ was used (Supplementary Fig. S8b), which was measured from the Dome C ice cores from Antarctica[42,43], interpolated to 1 kyr resolution. The sea level stack reconstructed from $\delta^{18}O$ data from ocean sediment

cores[44] was used in order to provide the GSL index used as the fifth driving parameter to the emulator (Supplementary Fig. S8c), which represents changes in global ice sheet volume, again interpolated to 1 kyr resolution. We equate a GSL index value of 0.0 with the modern ice sheet volume, and a GSL value of −125 m with the ice volume of the LGM. Prior to 800 ka, the emulator was run with orbital parameters calculated in the same way as in the 800 kyr simulations[2]. However, $CO_2$ and GSL were from different sources: (i) $CO_2$ was taken from a record of model-derived $CO_2$[21] that is consistent with global sea level reconstructions[45]; and (ii) GSL was taken from a 1-D ice sheet model to reconstruct sea level over the last 20 Myr[45]. Although there are more recent proxy-based estimates of $CO_2$[46], this cannot be used in our study as we require an orbitally-resolved record through the entire 2.58 Myr that captures the relationship between the $CO_2$ feedback and the direct orbital forcing. Moreover, this other $CO_2$ record is not orbitally resolved prior to the ice core record[46]. Both $CO_2$ and GSL were interpolated to 1 kyr resolution here, and these two records were spliced with the existing 800 kyr records (see above) during the overlapping 5 kyr (i.e., 795–800 ka).

### Proxy data compared with output from the emulator

The palaeoclimate data used to compare with the emulated SAT and PREC were taken from several different locations (Table 1 and Supplementary Fig. S1), and were selected due to their high temporal resolution and coverage of the Quaternary. For temperature, a quality control exercise was undertaken which resulted in the elimination of any records with: (i) a temporal resolution >5 kyr, which may not capture orbital-scale variability; (ii) a starting point >3 kyr BP, as the emulated anomalies were calculated using data between 0–3 kyr; and (iii) <500 kyr worth of data during the Quaternary, to allow a reasonable model-data comparison. This exercise yielded one record of reconstructed SAT, four records of reconstructed SST and three records of reconstructed precipitation:

- Reconstructed Antarctic SAT, derived from δD data from the Dome C ice core[42,43].
- Reconstructed SST data are based on a number of proxy methodologies, including the $U^K_{37}{}'$ index, Mg/Ca ratios from foraminiferal calcite, and the modern analogue technique (MAT), listed in Table 1.
- δ[18]O data measured from cave speleothems, which is taken to be a first-order approximation for intensification of East Asian monsoon rainfall. Records from three caves were used to provide full coverage of the last 640 kyr, namely the Hulu Cave in eastern China[47], the Sanbao Cave in central China[48], and the Dongge Cave in southern China[49].

Palaeo SST data is used for comparison with emulated SAT because multiple records exist from varying global locations, which cover hundreds of thousands of years at a sufficient resolution to capture orbital cycles. Outside regions of sea ice cover, the offset between SST and SAT at a single location is small compared with the local glacial-interglacial cyclicity. Each proxy data set was compared to the emulated mean annual SAT or PREC for the appropriate grid box for each site. Temperature and precipitation data are generally shown as anomalies compared to PI conditions at the different locations, taken as the difference with the average of the most recent 3 kyr in the case of SST, and the PI control simulation (i.e., 0 kyr) in the case of SAT and PREC.

### Step 5: Sensitivity simulations.

Once the above model-data validation process was complete, the final stage was to undertake a number of sensitivity simulations. Firstly to investigate the sensitivity to the various forcings of the emulated data at the proxy sites, through constant-driver simulations (Sensitivity of emulated results at proxy locations), secondly to identify a synthetic $CO_2$ forcing that results in the best-fit

of emulated climate to proxy data (Sensitivity of emulated results to CO2), and thirdly to separate out the dominant driving mechanisms from among the five input parameters, on a global scale (Sensitivity of emulated results to dominant driving input).

**Sensitivity of emulated results at proxy locations.** For the first investigation, a number of idealised sensitivity simulations, covering just the most recent 800 kyr, were carried out in order to investigate the sensitivity of the sites. To do this, instead of using actual realistic values for all five driving parameters (used as input when running the emulator), these were set as either (i) a constant average, (ii) constant PI, (iii) constant first year (because of missing data, the first year is not always the PI), (iv) constant LGM and (v) randomly generated values.

**Sensitivity of emulated results to $CO_2$.** For the second investigation, a synthetic timeseries of $CO_2$ was generated, by initially running the emulator multiple times using constant $CO_2$ as input, to provide a range of temperature values that sit above and below all possible proxy reconstructions. The other four input parameters were kept as naturally/realistically varying. Specifically, the emulator was run seven times for the full 2.58 Myr, with the $CO_2$ held constant in each simulation as multiples of the PI (280.4 ppm): 0.125 x, 0.25 x, 0.5 x, 1 x, 2 x, 4 x and 8 x (corresponding to 35.05, 70.1, 140.2, 280.4, 560.8, 1121.6 and 2243.2 ppm, respectively). Therefore, in these simulations, any variability in the resulting emulated temperature is associated with ice and/or orbital parameters only. Supplementary Fig. S4 in the Supplementary Material shows the resulting temperature from these simulations at one example location (in the central South Pacific), alongside the proxy data. These constant-$CO_2$ simulations (Supplementary Fig. S4a) were then used to estimate the $CO_2$ that would give the best fit of emulated temperature to proxy data. At every location and at every timeslice, the proxy temperature was identified, as was the emulated temperature immediately above and below and the constant $CO_2$ values used to create these temperatures. These values were then used to interpolate the exact $CO_2$ (initially in log space, and then converted back to actual $CO_2$) corresponding to each temperature. As a test, the emulator was then run again with these synthetic $CO_2$ curves, and the corresponding emulated temperatures were, as expected, identical to the proxy temperatures on which they were calculated (see Supplementary Fig. S4b). Finally, for each proxy record, the synthetic $CO_2$ was compared with $CO_2$ reconstructions, including the observed[42] and modelled[21] $CO_2$ used in the model-data comparison, as well as two other $CO_2$ records further back into the Quaternary: $CO_2$ at ODP 999, spanning 1088–1244 kyr[50] and $CO_2$ at ODP 668, spanning 1384–1535 kyr[51]. All $CO_2$ records were firstly interpolated to 1 kyr resolution to enable comparison with the emulated temperature and optimised $CO_2$ (see Supplementary Fig. S5). Note that although other $CO_2$ records exist even further back in time, such as for the Pliocene[52], these were not included here because they are not orbitally resolved over multiple successive glacial-interglacial cycles, and additionally, no SAT proxy data (which passed the above quality-control tests) were available for this time period.

**Sensitivity of emulated results to dominant driving input.** For the third investigation, the emulator was run with a single forcing or subset of the forcings being incrementally varied, following the multi-variate factorisation method[23], whilst the other forcings were held constant at PI values. For these, the nomenclature uses a binary system of 5 digits representing (in order) $CO_2$, obliquity, eccentricity, precession and ice, where 0 = constant PI values and 1 = varying values. Hence, $E_{00000}$ (the no drivers simulation) has all drivers set to constant PI values, $E_{10000}$ has varying $CO_2$ but everything else is set to constant PI values, and so on, up to $E_{11111}$ where all forcings are varying (the all drivers simulation). This gave a total of 32 simulations, capturing all possible combinations of forcings (see Supplementary Table S2 in the Supplementary

Material). To reiterate, the 32 simulations refer to the sensitivity simulations using the emulator, not the HadCM3 simulations used to train the emulator (of which there were 120 and 122 simulations for the interglacial and glacial emulators, respectively, as discussed above). For the factorisation, only the most recent 800 kyr were considered, as the input drivers (i.e., $CO_2$ and GSL) are most reliable during this time period.

## Metric for quantifying model performance

For the model-data comparison, idealised sensitivity simulations and factorisation, we generate a skill score for temperature (SAT and SST) of the emulator compared with the proxy data, using the non-dimensional arcsin Mielke measure, M[22]. This is the mean square error (MSE), non-dimensionalised by the variance[22]. For emulated values x and proxy data y:

$$M = (2/\pi) \arcsin\{1 - mse/[V_x + V_y + (G_x - G_y)^2]\}*1000 \qquad (1)$$

where V is the variance and G is the mean. For the model-data comparison, M is calculated between the emulated (using the $E_{IIIII}$ simulation) and proxy data at the same site (such as ODP 1012 versus proxy ODP 1012) and at every possible combination of sites (such as emulated ODP 1012 versus proxy ODP 1123). For the idealised sensitivity simulations, M is calculated between the emulated (using the $E_{IIIII}$ simulation) and the modified (to be constant) proxy data at the same site. For the linear and linear-sum/shared-interaction factorisation, M is calculated between the $E_{IIIII}$ simulation and all other simulations (i.e., treating the $E_{IIIII}$ simulation as observations). Whereas the model-data comparison figures presented here show anomalies (relative to the PI), the M scores were calculated using the emulated and proxy absolute values.

## Data availability

The data (used to create the results and figures in the main manuscript and Supplementary Information) generated in this study are too large to upload to a public repository; access can instead be obtained by contacting C.J.R.W. Likewise, all data required to train, build, optimise, test and run the emulator can be accessed by contacting C.J.R.W. and D.J.L., as there are too much gridded data to upload to a public repository.

## Code availability

Code for the wrapper of the GP package, as well as the various scripts to display/visualise the data, is available from https://github.com/cwilliams2020-new/Pleistocene_emulator[53]. Please note, however, that the code provided here is not entirely stand-alone, because in order to run it would require all of the HadCM3 simulations used to train the emulator (see above). These data can be accessed by contacting C.J.R.W. and D.J.L.

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

## Acknowledgements

The work described in this paper was carried out in the framework of a project funded by Posiva Oy, SKB and KAERI, funding C.J.R.W., N.S.L. and A.T.K. C.J.R.W. was also partly supported by the NERC - NSFGEO grant NE/Y001443/1 ('Pliocene Lessons for the Indian Ocean Dipole', PLIOD). Some of the earlier development work was carried out in the framework of a project funded by R.W.M., D.J.L. and X.R. also acknowledge support from NERC grant NE/V01823X/1 ('Solving the Oligocene icehouse conundrum'), and the NWS. C.J.R.W., D.J.L. and X.R. also acknowledge support from the TONIC grant (R102341-101). R.M.B. acknowledges support from NERC grant NE/L002531/1 ('The Southampton Partnership for Innovative Training of Future Investigators Researching the Environment', SPITFIRE). This work was carried out using the computational facilities of the Advanced Computing Research Centre, University of Bristol - http://www.bris.ac.uk/acrc/.

## Author contributions

C.J.R.W. conducted the emulator simulations, carried out the analysis, produced the figures, wrote the majority of the manuscript, and led the paper. N.S.L. initially set up the emulator and wrote some of the manuscript. A.T.K. provided technical assistance in running the emulator, and X.R. provided some of the figures. M.C. provided an earlier version of the emulator. D.A.R., A.K., M.T. and P.J.V. proofread the manuscript and provided edits. G.L.F., R.M.B. and E.L.M. provided the proxy reconstructions. D.J.L. contributed to some of the writing.

## Competing interests

The authors declare no competing interests.
