## [Transparent Peer Review file · Nature Communications]

The relative role of direct orbital forcing versus CO₂ and ice feedbacks on Quaternary climate

Corresponding Author: Dr Charles Williams

Version 0:

Reviewer comments:

Reviewer #1

(Remarks to the Author)

Williams et al 2024 Nat Comm. Review

This study uses two statistical climate (temperature and precipitation) emulators a stand-in for a forward climate model in order to do a simulation of the last 2.6 Million years while circumventing long integration times.

Overall, I do like the concept of model emulators to facilitate researchers parsing through large ensemble sets, filling in the blanks between known forward model integrations, mining through large parameter spaces, etc. It is highly likely that this technique will become a vital supplemental technique to forward models in the future.

However, although the application of this concept towards doing long paleoclimate simulations is somewhat novel (barring the heavily self-cited Lord et al 2017), the implementation here seems incomplete (could be done better).

Also—I do not understand why the authors (some of whom I SAW on the call on the PMIP WINGS seminar in fall 2023 actively listening to Axel Timmermann present) did not cite the Yun et al 2023 paper out of that group using the CESM1.2 model to do transient simulations of the exact same 3 Million years to present time period. <https://cp.copernicus.org/articles/19/1951/2023/cp-19-1951-2023.pdf> Its a pretty big glaring miss. And this is THE simulation to compare to now. (And maybe in another decade we'll have a few other models who have managed such a transient, but for now...)

First, the methods presented in the main paper are totally insufficient to understand what is going on. You must dig into the supplemental, or happen to catch in the discussion, to realize they are using HadCM3 – a CMIP3 class model – to do the training of their statistical emulator. (They should not use 'GCM' in the text—they should define HadCM3 is the GCM that they use, as opposed to taking ALL the various flavors of PMIP GCMs and training on those GCMs, the general term whereas they use HadCM3, a specific model.)

The emulator training should be described concisely in this paper so that it can stand alone. I found myself unable to read the text without going back and reading Lord 2017. After reading Lord 2017, I am still a little confused at a few things. Why are there only 4 parameters? What does it mean to do that with two distinct emulators? (Why isn't ice sheet extent and amoc response a 5th / n-th parameter?) Vegetation extent parameter? methane separate from co2 as some sort of proxy for veg extent perhaps? Lake extent parameter? Why did they only train for a global composite number? Why not regional? If you're going to train for a global composite number, why train on (?deep ocean?) temperature instead of training on what the archives actually are d18Oc or MgCa or UK37 or whatever. In the Lord 2017 paper, there is a thoughtful review exchange between Ganopolski and Crucifix. "In our case, as in most applications we have seen so far, the most important judgement is that the GCM response is smooth, but it does not need to be linear. Another important judgement is that the GCM internal variability is Gaussian." But is this true. Do ice sheets respond smoothly? Does AMOC? When you look at the emulators being done by folks at MIT, this part of the world they don't do so well. (<https://www.youtube.com/watch?v=PbcFWN5dtJc> 35:59) This project of course is much more expansive, but -BUT- I think it illustrates how tricky it IS to get AMOC and ice sheet responses right.

Next, is the parameter space of the Lord 2017 paper – which was made to look at high CO2 environments applicable here? (see next comment)

HadCM3 is a fine model to use for making an emulator—however, this model is also at this point relatively inexpensive to use. When they name drop on page 7, they admit to do the actual simulation using the real forward model would only take 280 days.

For a paleoclimate simulation, 9 months of integration really doesn't seem that bad. That seems pretty par for the course. For this study, they probably *should* go back and run many more simulations of HadCM3. The code repository says only 32 runs were used. That seems pretty tiny. And, the parameter space of the Lord 2017 paper, we later learn is not good enough, so they actually train another emulator for more ice climates. So this paper is really an amalgamation of two emulators. Which is a weird (not 100% sure defensible, especially as buried and not stress tested enough as this info is currently in the paper) place to be in if the times in between deglaciated and glaciated state is what they are trying to use the emulator to fill in.

Plus, there have been deglacial HadCM3 runs 'out there' for probably 20 years. Why not use them more. And there IS Ice6G from the Eemian onwards. Why NOT use lots of time slices there.

For the results pieces, the most interesting part – the Mid-Pleistocene-Transition doesn't really show up. Its disappointing that there are two end-state emulators and nothing in between. Isn't the MPT all about oscillating in a stable way in the in-between part? I don't quite follow why they are showing individual ODP cores when they are training to global numbers. Wouldn't some sort of global stack be better (actually, I think they should at the minimum follow the Yun23 example and use regional stacks)? Or, alternatively, train to all the various ODP and speleothem and ice records that are available. And then do tons of training/testing period exercises to quantify how putting in/out the various archives influences the answer.

It feels supremely hand-wavy to say that variability seems a combination of "CO2 forcing and ice sheet feedbacks" because they have emulated the former well while the latter is left in this weird zone between the two emulators.

In summary—I think this could be a great paper if the idea was fleshed out better, but I do not think my above concerns can be corrected with minor revisions. Its probably 6-12 months' work.

But the author should indeed persevere because this type of approach will become increasingly important, and it is a challenge to implement novel applications.

(Remarks on code availability)

They definitely have code there. I didn't look through it closely, but it has comments and is 'clean' enough.

(I'm a little surprised that R has the memory for this task.)

However,

When You open up one of the files, it has dependencies. These need to be packaged up better because I couldn't get the links to work. (Tarball all of these dependent, non public source R files)

```
source('C:/Users/cw18831/OneDrive - University of  
Bristol/Documents/Research/SKB/SKB_Alan_Code_DJL/PosivaSKB/PosivaSKB/Emulator/2015_Bristol_5D_v001/R/col_parula.R')  
source('C:/Users/cw18831/OneDrive - University of  
Bristol/Documents/Research/SKB/SKB_Alan_Code_DJL/PosivaSKB/PosivaSKB/Emulator/2015_Bristol_5D_v001/R/col_bwr.R')  
source('C:/Users/cw18831/OneDrive - University of  
Bristol/Documents/Research/SKB/SKB_Alan_Code_DJL/PosivaSKB/PosivaSKB/Emulator/2015_Bristol_5D_v001/R/col_wr.R')
```

Reviewer #2

(Remarks to the Author)

Review of "The relative role of orbital forcing, CO2 and ice sheet feedbacks on Quaternary climate" by Williams et al.

The manuscript of Williams et al. shows a long simulation of glacial and interglacial cycles over the Quaternary. For that purpose, they use an emulator calibrated from a coupled model. They show that CO2 and ice sheet feedbacks are the main driver of the temperature variations over those cycles and that the orbital forcing effect is minimal.

Overall, the manuscript of Williams et al. is truly fascinating, well-written and it was a pleasure to review it. However, I do think some of the most interesting features are the least highlighted. I also have a few comments to improve the clarity of the manuscript which in some places is not so easy to understand, in particular for someone who is not into the field.

Major comments:

L93-94-95 (and in the whole manuscript in general): I think there is a big lack of explanations regarding what an emulator is. I have never heard of them before, and without looking at the methods, I still had absolutely no idea what it is and how it works by the end of the paper. I understand that Nature's papers have a structure such that methods will be after the core of the manuscript, but I think a few lines need to be added in the main text to explain more in details what is an emulator. Does it have any kind of physics? Is it some kind of extrapolation method? Calling it just "a statistical model which estimate uncertainty" is just vague and means everything and nothing at the same time. It is very difficult for a reader to constantly jump back and forth between the text and the methods while keeping motivated to continue reading. For instance, some of the sentences around L387 were quite helpful for me to understand.

L166-167-168 (and in the whole manuscript in general): This whole paragraph is crazy to believe and I found it to be the most interesting part of the paper. While I understand the importance of simulating glacial/interglacial cycles, I could not say I was surprised at all CO2 and ice sheet feedbacks are the main drivers. The main topic that amazed me in this paper is the use of a method, which seems like a massive step forward in climate modelling to simulate those cycles. After reading this, I was just

asking myself "Why do we bother using GCMs then?". I feel that the efficacy of this emulator and its potential use in any field in paleoclimate modelling is as important, if not more, than the other results of this paper. Yet, I think the authors did not discuss this aspect so much. I feel the paper's highlight is really this role of CO₂ and ice sheets, while the authors basically showed that an emulator could replace a GCM in any kind of sensitivity tests and many complex situations, all of that while saving an enormous amount of resources. So on one hand I have the role of CO₂ feedback in glacial cycles, and on the other hand a critical tool that could be a revolution in paleoclimate modelling...

L262: Since orbital forcing seems to contribute so little to temperature signal, does that imply constraining orbital forcing in any PMIP runs is pointless? PlioMIP3 is exploring sensitivity runs with different orbital values. Does that make any sense? Also, is this result strongly model dependent?

L322: Are there any kind of vegetation feedbacks in the emulator which could also influence the temperature signal?

L333-334: I wish the authors would talk more on this point, because from what I understood from the rest of the paper, this aspect does not seem very difficult to test considering the performance of an emulator. I think one needs some kind of large ensembles, but those large ensembles already exist. Is the issue that you need large ensembles in both glacial and interglacial conditions?

Minor comments:

L83: Too many commas? "A different, but complementary, approach is..."

Fig.1: I am not sure about Nature's requirements regarding figure but if the figure was in PDF it would really improve it... The captions talk about "Light blue error bands" but with this poor figure quality it is impossible to see much.

L152: Maybe a very brief explanation of what an M score is in this caption is required (I know it is in the method but it is the same issue as the lack of explanation regarding an emulator in the core of the manuscript)

L206: Typo "underestimate the show the correct..."

(Remarks on code availability)

Version 1:

Reviewer comments:

Reviewer #1

(Remarks to the Author)
Williams et al 2024 Nature
Second phase review

The manuscript is much improved from what was submitted previously.

However, I reiterate here as I have to the editor that I am an emulator adjacent scientist. This manuscript absolutely must be evaluated by a researcher who actively uses emulators and is familiar with the quickly changing field of AI/ML in climate science. Although I work with colleagues who are experts in emulators and feel very familiar with their use and applications, I do not use them in my own research and thus do not feel qualified to comment on the particulars of the technique applied here.

As in my previous comments, there are many important feedbacks within the earth's climate in modulating the climate response. The PMIP/CMIP literature acknowledges this fact with its 'spur' experiments: lgm lgm+dust midholocene midholocene+veg (plus methane lakes etc) ... ismip experiment ... volmip experiment ... nahosmip ... In the first paragraph of the intro, the things your study can address are listed, but the things you did not consider are not discussed. Be clear to set up the limitations and expectations here and realistic about what the implication of these exclusions. These caveats belong up top, in the first paragraph or two, and you can even set up a nice resolution of the outstanding question- whether the exclusion meant you missed out on important features in the transient. It is especially disappointing to have 3 of the forcings being the orbital parameters—which are then determined to be unimportant using the M score (which isn't quite believable). And it isn't clear to me that using the parameter coefficients of orbital forcing versus some derived quantity for insolation is correct. More details on orbital forcing are required for such a ringer conclusion (contrary to pretty much all the paleoclimate literature out there) about orbital forcing not being the driver and most important forcing. And as a set, since CO₂ is how orbital forcing gets parsed through the carbon cycle and ice sheets are how that gets parsed into the cryosphere, perhaps it IS better to use other derived quantities like vegetation or dust or 1st derivative of sea level (ice melt) or similar as the forcings instead of the raw orbital parameters.

The methods section is much better now. But the main text readability isn't that great. Too much in the weeds about certain things without enough detail to follow the argument for others (details below). Stylistically, I think the main text needs a strong editing hand.

I also did ask that the authors make plenty of comparisons to the Yun et al 2023 simulation which is the most similar to their efforts. They have done that—but that is not to say that this should be to the exclusion of other comparisons and efforts.

There is an implicit question through the text that needs explicit address:: hysteresis. That is to say, does the climate response

depend on the initial state. This question comes up in the discussion about the relative merits of the speeding-up technique used in the Yun 2023 manuscript versus the emulator. Does the emulator include a phase delayed temperature (CO₂ et al) forcing field or anything else to include initial state? Or is it judging each climate state independently.

It STILL appears that there are two separate emulators (glacial and interglacial) that are stitched together. Is it any wonder that the main finding of ice sheet is most important is arrived upon when it's baked in to the process? I remain concerned, even after reading the responses and rewritten methods section, that there are two distinct emulators. Without this continuum, what chance is there to capture changes at the mid-pleistocene transition? Authors aren't using a linear regression that might suffer from assigning warming of similar magnitude to cooling from G to IG states. That's the point of this method. I am not convinced about the use of two separate emulators. What happens if they use just one. L529 'steps were taken' -> this is the methods section to describe those steps.

I still remain concerned about emulating AMOC as per the first review. I do not see this addressed in the main text, not relegated to a mention deep-deep in the methods L542. AMOC remains one of the confounding factors for folks working on climate model emulators and it is odd to not explicitly address this piece.

Major revisions are still needed, although somewhat less major than last round. This emulator exercise is important for the paleoclimate community, although I am not convinced the present work is complete enough as presently written (and possibly done for the forcing comments above).

L86: This is where you should explicitly state that this Yun23 type of simulation assumes that the model simulation is relatively insensitive to initial conditions – you need to also discuss your own initial conditions. Was the training HadCM3 mindful of initial conditions? (Did you start your Pliocene simulations from an existing Miocene simulation or was it initialized from a default configuration.) If it was not, then you also have this same potential <lack of> initial condition bias built in.

Hysteresis (aka tipmip and the like) is a major point of concern for future comparisons. iow: I would frame this comparison a bit differently. It does nothing to degrade your work by having another group achieve a similar result—rather it gives an excellent basis for comparison with strengths and weaknesses of your (their) approach, (in)consistencies, etc.

L103-5: "simulate" => "interpolate" this is what you are doing, you are not simulating. Emulators interpolate points determined by data or physics-based models, and totally dependent on the training data "points". The word "emulator" denotes the relatively new application of new AI/ML techniques instead of traditional to the statistical models reproduction—please introduce this concept in this paragraph.

L119: "proxy" => "past climate" using "proxy" as a noun becomes ambiguous as you cross fields.

L120: "physics-based", more complex GCM...

L121: the number of simulations used in the training data is related to the number of parameters explored – use the specific language. What did you use to properly sample the phase-space? Latin Hypercube? Something else? Introduce here. How many simulations are required per parameter?

L126-128: In reading this section, the word parameter feels muddled with the word parameter as it is used in perturbed physics ensembles—those take on the PPE acronym. Perhaps there is a use-specific acronym? Boundary-Condition-Parameter (BCP)? This technique has been applied to sampling the parameter space used within the model physics (eg doi:10.22541/essoar.172745119.96698579/v1), to interpolate between future earth scenarios (eg <https://arxiv.org/html/2404.13227v2>)

L141: This stat of '5 minutes to run XX years' will not age well and there is not enough info about the emulator in-line to know how complex it is (Also- is the emulator full atmosphere 3d, ocean 3d?, land surface 3d?). Framing in comparison to the amount of time required to run the full model is sufficient.

Further, the emulator is solving each state as if it is totally independent (my take after reading the methods but this should be spelled out briefly in the text) — which is equivalent to doing time slices. Authors use 1kyr resolution, which is equivalent to doing 2580 time slice simulations. The emulator was trained on 182, 500 year time slices. Since they provide 120 years / day as the length scale of these simulations, these training simulations took <5 days. If the runs were done not overlapping, that is 29 years. Running the ensemble at a more typical few dozen at a time, that would be a run time on order of a year or so. In any case, there is too much in the weeds on this run time justification in the main text and not enough details on the emulator in the main text.

L151: Does HadCM3 really output subdaily diagnostics and then average those into monthly averages? That is an interesting choice. I would say most groups do not do that I/O step because I/O tends to slowdown the simulation A LOT because you are stopping and writing to the disk and depending on the filesystem that can be rather slow. Many groups keep up with tracers in memory at the normal timestep (20-30 min), and then only output at the temporal resolution required, vastly decreasing I/O. Are the authors positive that HadCM3 does not work that way? The next few sentences are too much in the weeds for main text and should be moved to the methods or supplemental.

L189: The discussion of figure 1 is too qualitative. Did you plot and calculate T_emulator versus T_proxy? The figure 1 plot here takes up too much space and yet the curves are really too tiny and delicate to 'tell' what the authors are talking about.

There also needs to be a context of traditional 'model' built in. If you simply regress your forcing fields on (say) the 3d SST field, how does this statistic compare with the emulator's? $T_{regress}$ versus T_{proxy} ? For the d18O_{ice} record, how do your results stack up against the EPICA Grand Challenge results (where simple and fast box models were applied to link CO₂ and temperature for Antarctic ice core).

With these other methods, do you have the same bias periods that you lay out on L193?

On L211: 4degC is FAR too little cooling for last glacial maximums. This magnitude should be set by data assimilation methods eg <https://doi.org/10.1038/s41586-021-03984-4> and in context of PMIP3/PMIP4 LGM models.

L 204/701: You talk about the interpolated CO₂ up to 800k in this study and then punt to the methods. Probably better to put concisely enough in the text to follow what you've done (we interpolated 10k resolution to 1k resolution etc). You use an outdated source for CO₂ prior to 800k. I recommend <https://doi.org/10.1126/science.adi5177> which is the recent Dec2023 GenCO₂PIP consensus paper.

L217: Are you saying that you have the same as the Yun paper or the same variability. It isn't clear.

L221: Again here, this would show up better in a scatter plot to augment figure 1 as the highest and lowest values having the greatest error.

L232: Can you say the interpretation something like "G-IG CO₂ would need to be 1.5 times larger to match the proxy G-IG temperatures (Figure S6)" instead of this hand waving that requires digging into the supplemental. After looking at Figure S6, I am not entirely sure what you've done. Do you not have actual CO₂ in 800k to 1.1k? Are you comparing optimized CO₂ to actual CO₂? Can you put numbers on this?

L239: OR your mismatch could be that you have not included enough forcing fields. Like vegetation or etc written above.

L243: MAP is defined above, but it is an uncommon shorthand and makes the paper hard to read. SAT, in contrast, is a common acronym. Why not use PREC to improve readability?

Also, Chinese speleothem d18O are not MAP. They are changes in d18O that memorializes monsoon strength.

L262: 'all drivers' simulation (E11111) Each subscript means forcing on or off? Which is which? Is this spelled out somewhere? Although this shorthand likely works great in the lab, it is hard to read.

L265: as above, some sort of basic regression model comparison/context should be added.

L269: I don't follow what this means. Constant climate signal of what? Do you mean "equilibrium" climate experiment like PMIP? If that's what you mean, what were their scores?

L289: You cannot punt folks to the supplemental text and then descend into unfamiliar jargon like "where each driving component is averaged over all 120 pathways" I have no idea what you are talking about. The supplemental should AUGMENT the main text, not render it unreadable. When you read in the supplemental the full suite of runs—decode them here otherwise readers have to go back and dig into the methods (as per comment above about E11111), and then dig back into the original point in the paper. This isn't readable. Please add a supplemental table what E10000, E01000... E00001 mean.

L293: If you get a result that the impact of the orbital forcing on M scores is negligible—perhaps this is an indicator that the orbital 'forcing' the way you've implemented it isn't quite right. It is the impact that these have on insolation seasonality that is important. And it seems weird to have eccentricity separate from precession as two separate forcings.

What is the M score of the orbital parameters to CO₂ and ice sheet volume? These need to be plugged in to a function to yield insolation. Does the M score technique capture that?

Perhaps you should use a variation of the original specmap methods (insolation at 65N) Perhaps max-min insolation at mid latitudes of northern and southern hemisphere (or something to indicate seasonality magnitude) plus a baseline total insolation at (say) 65N?

Given that the emulator is so inexpensive to run, it would seem that you could experiment with other forcing parameters.

Or perhaps this is an indication that the orbital forcing is already built in to the CO₂ and ice volume signal so well that they are redundant? And maybe these 3 parameters should be swapped out with vegetation. Vegetation could be distilled down to the midHolocene pmip variations (fraction of 'green sahara' and shrubs at high latitudes). Or maybe other things like aerosol loading might be used as a forcing (as CO₂, easier done 800k to present than before). Or perhaps use the first derivative of the sea level (something like amount of ice melt in NH and SH).

In short, do not think this paragraph L297 with the orbital parameters causing <5% of the signal is defensible as written and goes against 35 years of prevailing thought.

L346: Why are there only modern and lgm ice? I know that there are HadCM3 deglacial experiments. I realize that I made this comment before, and there is a response to reviewer, and yet I am not satisfied given that the lack of ice variability is one of the

issues discussed in the discussion.

L642: You should use BOTH the updated Hargreaves/Annan estimates (Renoult) generally on the low-end of accepted across the community and the Osman/Tierney estimates generally on the higher end of accepted across the community from data assimilation methods here instead of this older work.

(Remarks on code availability)

No I didn't look through the code this time.

Reviewer #2

(Remarks to the Author)

No more comments (all comments have been answered in the new version of the paper)

(Remarks on code availability)

Reviewer #3

(Remarks to the Author)

The idea of a paleoclimate emulator is a good one. These simulations are so long and expensive that we already (often) use pared down climate models, and this is taking it another step to pare things down even further, which is a worthy goal. The emulator seems to be well constructed and carefully validated. In general, I really like this manuscript. However, I have a question about whether it fits the scope of the journal.

Where I struggle with this paper is that it seems to be focused on describing and validating the emulator's performance, which is what I might expect for a more methods-focused journal. What I think is lacking are insights that the emulator can provide. Examples could include figuring out what processes are important to represent in models for understanding paleoclimate variability or revealing drivers of past changes. The authors do point out that CO2 concentration and ice are more important than the inputs corresponding to orbital parameters, but given the way the results are presented and caveated, I can't tell whether those results are new or surprising, or even whether they are simply a product of what was done.

My hope is that these issues can be taken care of via wording and do not require much new analysis. The paper is quite well done, and the tool the authors have produced is quite valuable.

The authors also did an excellent job of responding to the previous review comments, some of which were quite challenging.

(Remarks on code availability)

The other reviewers focused on the code review, so I didn't do this part.

Version 2:

Reviewer comments:

Reviewer #1

(Remarks to the Author)

Review of Williams et al 2025

This paper is greatly improved from previous versions. I do not think I will need to see this again before the editor makes a decision.

I'm still not satisfied with two stitched together emulators being used instead of one. It makes the very interesting and good work seem less so because the answer gets baked in to the training.

"The approach of having two separate emulators was taken to ensure that there was no 'leakage' of state-specific climate variations across different climate states (see Supplementary Material, Figure S1e). For example, under glacial conditions development of the Laurentide ice sheet is accompanied by strong cooling over northern North America. However, this does not mean that during interglacial conditions there should be a strong warming in the same region, since the data are presented as an anomaly from interglacial present-day conditions (where no ice sheet is present)."

Actually, the change in height alone does mean there is rightfully a huge amount of warming in this area. In anomaly space this does look odd—it just does. If you want to get rid of that, try using vertical lapse rate to apply a correction.

If the authors used a single emulator – show it. It makes no sense the explanation that there is leakage over North America and then differences between the two emerge on the opposite side of the world "When compared to proxy data at the five sites presented in Figure 1, the results showed no difference in SST but higher variability for SAT over Antarctica (not shown)". Show both emulators.

The “M score” is not described adequately for readability in the text. In the main text, just a few short words like “M-score (Arcsin-Mielke measure M) quantifies the goodness of fit of the emulator to the proxy climate archives such that where no skill is 0 and maximum skill is 1000 (see methods)” will allow the reader to keep reading without “flipping” to the back page.

The authors insist on using the 2013 Annan and Hargreaves extremely low LGM number which no one believed at the time, and not even they believe today, as memorialized here

Annan, J. D., Hargreaves, J. C., and Mauritsen, T.: A new global surface temperature reconstruction for the Last Glacial Maximum, *Clim. Past*, 18, 1883–1896, <https://doi.org/10.5194/cp-18-1883-2022>, 2022.

“Abstract

We present a new reconstruction of surface air temperature and sea surface temperature for the Last Glacial Maximum. The method blends model fields and sparse proxy-based point estimates through a data assimilation approach. Our reconstruction updates that of Annan and Hargreaves (2013), using the full range of general circulation model (GCM) simulations which contributed to three generations of the PMIP database, three major compilations of gridded sea surface temperature (SST) and surface air temperature (SAT) estimates from proxy data, and an improved methodology based on an ensemble Kalman filter. Our reconstruction has a global annual mean surface air temperature anomaly of -4.5°C relative to the pre-industrial climate. This is slightly colder than the previous estimate of Annan and Hargreaves (2013), with an upwards revision on the uncertainty due to different methodological assumptions. It is, however, substantially less cold than the recent reconstruction of Tierney et al. (2020). We show that the main reason for this discrepancy is in the choice of prior. We recommend the use of the multi-model ensemble of opportunity as potentially offering a credible prior, but it is important that the range of models included in the PMIP ensembles represent the main sources of uncertainty as realistically and comprehensively as practicable if they are to be used in this way.”

Just take Annan and Hargreaves 2013 out. Replace it with something folks DO believe. (And many people still believe that -4.5°C is on the low end.)

The LGM reconstruction of Ice5G is not a very good one. When it came out, no one believed it either – particularly people studying the Laurentide Ice Sheet. They did not believe it because it pulled all of the downsizing ice from the Fenno-Scandian and Antarctic and piled it high on the Laurentide. Unbelievably so. Ice 4G and 6G are perhaps defensible, but 5G is not. The climate regimes by such a large ice mountain are just different from those without it.

The paper reads much better. And I’m glad the authors took the suggestion to cite more works beyond Yun 2023. But now the discussion slogs through Erb 2015 and it starts to read like a review instead of a nature article. There are many more models and simulations beyond Erb and Yun that have explored the ice sheet height and sensitivity of the climate.

I thank the authors for their clear explanation

“CO₂, obliquity, eccentricity, precession

and ice, where 0 = constant PI values and 1 = varying values. Hence E00000 (the ‘no drivers’ simulation) has all drivers set to constant PI values, E10000 has varying CO₂ but everything else is set to constant PI values, and so on, up to E11111 where all forcings are varying (the ‘all drivers’ simulation).”

In addition to the (correct) explanation that eccentricity and precession are convolved, why have 5 subscripts instead of 3 (let the last three be ‘orbital’)? The authors state plainly that eccentricity is not tested without precession. Is obliquity tested separately?

If obliquity IS tested separately, it would be very interesting to know if obliquity is more important than precession/eccentricity as obliquity has been proposed as the cause of glacial cycles (Huybers etc). It would also be interesting to know if this was not the result.

The presentation of figure 1 is improved—but I still find the lines too delicate – are the barely visible fuzzy nature the uncertainty? It is tricky to have a multipanel plot with the much larger Antarctic temperature shifts and the b-e panels then scaled way too zoomed out. There is still much white space and what feels like default subplots. This can still be done better – literally most paleoclimate folks do this with having no extra x-axes, moving the curvy lines physically closer together, alternating y1 and y2 axes labels, and expanding the smaller magnitude to be visible (just make sure to note clearly that this is what you’ve done).

Figure 3. Figure S2. Figure S4. Figure S8. Use a projection like Robinson (etc) that doesn’t blow up the poles so much. I really have no idea why all the scripting languages let awful projections be the default.

(Remarks on code availability)

Reviewer #3

(Remarks to the Author)

The authors have done a nice job, and I have no additional comments. Reviewer #1 did raise some important questions that seem to be satisfactorily answered, but I’ll happily defer to them.

(Remarks on code availability)

**Responses to reviewers following submission of manuscript
“The relative role of orbital forcing, CO₂ and ice sheet feedbacks on
Quaternary climate” by Williams *et al.***

REVIEWER COMMENTS

We extend our sincere appreciation to both Reviewers for a thorough examination of our manuscript. Here, we address their suggestions, comment-by-comment. In the below, reviewers' comments are blue and in a smaller font, and our corresponding response follows in a standard font. Line numbers in the following refer to the manuscript.

Reviewer #1 (Remarks to the Author):

Williams et al 2024 Nat Comm. Review

This study uses two statistical climate (temperature and precipitation) emulators a stand-in for a forward climate model in order to do a simulation of the last 2.6 Million years while circumventing long integration times.

Overall, I do like the concept of model emulators to facilitate researchers parsing through large ensemble sets, filling in the blanks between known forward model integrations, mining through large parameter spaces, etc. It is highly likely that this technique will become a vital supplemental technique to forward models in the future.

We thank the reviewer for their comments.

However, although the application of this concept towards doing long paleoclimate simulations is somewhat novel (barring the heavily self-cited Lord et al 2017), the implementation here seems incomplete (could be done better).

The methods section, and associated description of the implementation, has been rewritten (see Methods, lines 453-814) and comments below.

Also—I do not understand why the authors (some of whom I SAW on the call on the PMIP WINGS seminar in fall 2023 actively listening to Axel Timmermann present) did not cite the Yun et al 2023 paper out of that group using the CESM1.2 model to do transient simulations of the exact same 3 Million years to present time period. <https://cp.copernicus.org/articles/19/1951/2023/cp-19-1951-2023.pdf> Its a pretty big glaring miss. And this is THE simulation to compare to now. (And maybe in another decade we'll have a few other models who have managed such a transient, but for now...)

This has been done, discussing the Yun et al. 2023 paper extensively in the Introduction (lines 83-95), Results (lines 206-219) and Discussion (lines 313-330).

First, the methods presented in the main paper are totally insufficient to understand what is going on. You must dig into the supplemental, or happen to catch in the discussion, to realize they are using HadCM3 – a CMIP3 class model – to do the training of their statistical emulator. (They should not use ‘GCM’ in the text—they should define HadCM3 is the GCM that they use, as opposed to taking ALL the various flavors of PMIP GCMs and training on those GCMs, the general term whereas they use HadCM3, a specific model.)

The methods section has been rewritten (see Methods, lines 453-814), such that it is now understandable as a stand-alone document. Terminology and definitions have been refined throughout the Methods section.

The emulator training should be described concisely in this paper so that it can stand alone.

See above response.

I found myself unable to read the text without going back and reading Lord 2017. After reading Lord 2017, I am still a little confused at a few things. Why are there only 4 parameters? What does it mean to do that with two distinct emulators? (Why isn't ice sheet extent and amoc response a 5th / n-th parameter?) Vegetation extent parameter? methane separate from co2 as some sort of proxy for veg extent perhaps? Lake extent parameter? Why did they only train for a global composite number? Why not regional? If you're going to train for a global composite number, why train on (?deep ocean?) temperature instead of training on what the archives actually are d18Oc or MgCa or UK37 or whatever.

Firstly, see above response about the methodology being rewritten, so that it is now a stand-alone document and understandable without reading Lord *et al.* (2017). This now explains the 5 input parameters (see Methods, lines 477-483), and why there are two distinct emulators (lines 485-546). Secondly, as discussed in the Methods (lines 569-574), the parameters mentioned in the comment above (e.g. vegetation extent) are not included in the emulator but are accounted for by the underlying GCM simulations used to train the emulator. Thirdly, the emulator is not trained on a global composite number or regional data, but rather it is trained on the spatial patterns from the GCM simulations i.e. global gridded data (lines 563-567).

In the Lord 2017 paper, there is a thoughtful review exchange between Ganopolski and Crucifix. "In our case, as in most applications we have seen so far, the most important judgement is that the GCM response is smooth, but it does not need to be linear. Another important judgement is that the GCM internal variability is Gaussian." But is this true. Do ice sheets respond smoothly? Does AMOC? When you look at the emulators being done by folks at MIT, this part of the world they don't do so well. (<https://www.youtube.com/watch?v=PbcFWN5dtJc> 35:59) This project of course is much more expansive, but - BUT- I think it illustrates how tricky it IS to get AMOC and ice sheet responses right.

This question is addressed by the (rewritten) evaluation of the emulator, discussed in the Methods (lines 615-628). The training GCM data includes non-linearities such as changes in the AMOC. The leave-one-out the analysis demonstrates that the emulator can capture these non-linearities.

Next, is the parameter space of the Lord 2017 paper – which was made to look at high CO2 environments applicable here? (see next comment)

The input parameter space has now been explained more clearly (see Methods, lines 477-483).

HadCM3 is a fine model to use for making an emulator—however, this model is also at this point relatively inexpensive to use. When they name drop on page 7, they admit to do the actual simulation using the real forward model would only take 280 days. For a paleoclimate simulation, 9 months of integration really doesn't seem that bad. That seems pretty par for the course.

We thank the reviewer for highlighting this error. This is now been corrected (see Results, lines 140-163), showing that actually it would take many decades to run HadCM3 for the equivalent amount of years.

For this study, they probably *should* go back and run many more simulations of HadCM3. The code repository says only 32 runs were used. That seems pretty tiny. And, the parameter space of the Lord 2017 paper, we later learn is not good enough, so they actually train another emulator for more ice climates. So this paper is really an amalgamation of two emulators. Which is a weird (not 100% sure defensible, especially as buried and not stress tested enough as this info is currently in the paper) place to be in if the times in between deglaciated and glaciated state is what they are trying to use the emulator to fill in.

Rather than 32 runs being used (in the comment above), there were actually 182 GCM simulations (of 500 years each, giving a total of 91 kyr) that were used to train the emulator.

The 32 relates to the number of emulator simulations carried out as part of the multi-variate factorisation, not the number of GCM simulations used to train/calibrate the emulator (see Methods, lines 781-794).

Plus, there have been deglacial HadCM3 runs 'out there' for probably 20 years. Why not use them more. And there IS Ice6G from the Eemian onwards. Why NOT use lots of time slices there.

See above comment. We acknowledge that there are more model data available but, as discussed in the Methods (lines 548-551), the GCM simulations used here, taken together, capture the full range from interglacial states to full glacial conditions, and the simulations of Singarayer and Valdes (2010) that are used as part of the training data do include the deglaciation.

For the results pieces, the most interesting part – the Mid-Pleistocene-Transition doesn't really show up. Its disappointing that there are two end-state emulators and nothing in between. Isn't the MPT all about oscillating in a stable way in the in-between part? I don't quite follow why they are showing individual ODP cores when they are training to global numbers. Wouldn't some sort of global stack be better (actually, I think they should at the minimum follow the Yun23 example and use regional stacks)? Or, alternatively, train to all the various ODP and speleothem and ice records that are available. And then do tons of training/testing period exercises to quantify how putting in/out the various archives influences the answer.

Firstly, a discussion about why the MPT doesn't obviously show up has now been included (see Discussion, lines 332-340). Secondly, we are not training the emulator on global numbers but rather on globally gridded data i.e. capturing the spatial patterns from the GCM; therefore a comparison of the emulator results with individual ODP cores is appropriate. This is discussed more clearly in the Methods (lines 563-567).

It feels supremely hand-wavy to say that variability seems a combination of "CO2 forcing and ice sheet feedbacks" because they have emulated the former well while the latter is left in this weird zone between the two emulators.

Neither CO₂ nor ice sheets (as represented by global sea level) are produced by the emulator, but rather both are used as inputs to the emulator and are based on proxy data. Therefore, given that the final part of the Results section discusses the output of the emulator (i.e. based on temperature only), and identifying which of the 5 input parameters are providing the largest signal (via the multi-variate factorisation), we consider this observation to be appropriate.

In summary—I think this could be a great paper if the idea was fleshed out better, but I do not think my above concerns can be corrected with minor revisions. Its probably 6-12 months' work.

But the author should indeed persevere because this type of approach will become increasingly important, and it is a challenge to implement novel applications.

We thank the reviewer for their comments, and apologise for any confusion caused by a disjointed and unclear Methods section. We very much hope that the methodology is now much clearer to understand, and without referencing multiple other sources.

Reviewer #1 (Remarks on code availability):

They definitely have code there. I didn't look through it closely, but it has comments and is 'clean' enough.

(I'm a little surprised that R has the memory for this task.)

However,

When You open up one of the files, it has dependencies. These need to be packaged up better because I couldn't get the links to work. (Tarball all of these dependent, non public source R files)

```
source('C:/Users/cw18831/OneDrive - University of  
Bristol/Documents/Research/SKB/SKB_Alan_Code_DJL/PosivaSKB/PosivaSKB/Emulator/2015_Bristol_5D_  
v001/R/col_parula.R')  
source('C:/Users/cw18831/OneDrive - University of  
Bristol/Documents/Research/SKB/SKB_Alan_Code_DJL/PosivaSKB/PosivaSKB/Emulator/2015_Bristol_5D_  
v001/R/col_bwr.R')  
source('C:/Users/cw18831/OneDrive - University of  
Bristol/Documents/Research/SKB/SKB_Alan_Code_DJL/PosivaSKB/PosivaSKB/Emulator/2015_Bristol_5D_  
v001/R/col_wr.R')
```

The dependencies mentioned above are not standard R routines, publicly available from any R repository (e.g. <https://www.r-project.org/>). All R routines that are personal to this work have been uploaded.

Reviewer #2 (Remarks to the Author):

Review of “The relative role of orbital forcing, CO₂ and ice sheet feedbacks on Quaternary climate” by Williams et al.

The manuscript of Williams et al. shows a long simulation of glacial and interglacial cycles over the Quaternary. For that purpose, they use an emulator calibrated from a coupled model. They show that CO₂ and ice sheet feedbacks are the main driver of the temperature variations over those cycles and that the orbital forcing effect is minimal.

Overall, the manuscript of Williams et al. is truly fascinating, well-written and it was a pleasure to review it. However, I do think some of the most interesting features are the least highlighted. I also have a few comments to improve the clarity of the manuscript which in some places is not so easy to understand, in particular for someone who is not into the field.

We thank the reviewer for their comments.

Major comments:

L93-94-95 (and in the whole manuscript in general): I think there is a big lack of explanations regarding what an emulator is. I have never heard of them before, and without looking at the methods, I still had absolutely no idea what it is and how it works by the end of the paper. I understand that Nature's papers have a structure such that methods will be after the core of the manuscript, but I think a few lines need to be added in the main text to explain more in details what is an emulator. Does it have any kind of physics? Is it some kind of extrapolation method? Calling it just “a statistical model which estimate uncertainty” is just vague and means everything and nothing at the same time. It is very difficult for a reader to constantly jump back and forth between the text and the methods while keeping motivated to continue reading. For instance, some of the sentences around L387 were quite helpful for me to understand.

The methods section has been rewritten (see Methods, lines 453-814). In particular, to answer the above questions about what exactly the emulator is and whether it contains any physics, please see Methods (lines 569-574), in combination with the rest of that section.

L166-167-168 (and in the whole manuscript in general): This whole paragraph is crazy to believe and I found it to be the most interesting part of the paper. While I understand the importance of simulating glacial/interglacial cycles, I could not say I was surprised at all CO₂ and ice sheet feedbacks are the main drivers. The main topic that amazed me in this paper is the use of a method, which seems like a massive step forward in climate modelling to simulate those cycles. After reading this, I was just asking myself “Why do we bother using GCMs then?”. I feel that the efficacy of this emulator and its potential use in any field in paleoclimate modelling is as important, if not more, than the other results of this paper. Yet, I think the authors did not discuss this aspect so

much. I feel the paper's highlight is really this role of CO2 and ice sheets, while the authors basically showed that an emulator could replace a GCM in any kind of sensitivity tests and many complex situations, all of that while saving an enormous amount of resources. So on one hand I have the role of CO2 feedback in glacial cycles, and on the other hand a critical tool that could be a revolution in paleoclimate modelling...

See above comment about the Methods rewrite, and in particular the Results (lines 140-163) where we discuss in more detail the efficiency of the emulator compared to a GCM.

L262: Since orbital forcing seems to contribute so little to temperature signal, does that imply constraining orbital forcing in any PMIP runs is pointless? PlioMIP3 is exploring sensitivity runs with different orbital values. Does that make any sense? Also, is this result strongly model dependent?

Some text on the above question has now been included in the Discussion (lines 382-390). But in short, no, constraining orbital forcing in PMIP runs is not pointless, because here we are looking at annual means only, for which orbital changes only have a small impact. If we were to look at seasonal changes, a separate emulator would be needed and then the orbital component would have more of an impact.

L322: Are there any kind of vegetation feedbacks in the emulator which could also influence the temperature signal?

This is now more clearly discussed in the Methods (lines 569-574). But in short, no, parameters such as vegetation are not included in the emulator but are accounted for by the underlying GCM simulations used to train the emulator.

L333-334: I wish the authors would talk more on this point, because from what I understood from the rest of the paper, this aspect does not seem very difficult to test considering the performance of an emulator. I think one needs some kind of large ensembles, but those large ensembles already exist. Is the issue that you need large ensembles in both glacial and interglacial conditions?

This has now been done as part of the methodology rewrite (see Methods, lines 453-814), in particular lines 485-551 where we discuss the different ensembles going into the glacial and interglacial versions of the emulator.

Minor comments:

L83: Too many commas? "A different, but complementary, approach is..."

This has been addressed.

Fig.1: I am not sure about Nature's requirements regarding figure but if the figure was in PDF it would really improve it... The captions talk about "Light blue error bands" but with this poor figure quality it is impossible to see much.

The caption here has been modified and the light blue error bands have been removed, as they are not relevant to the study.

L152: Maybe a very brief explanation of what an M score is in this caption is required (I know it is in the method but it is the same issue as the lack of explanation regarding an emulator in the core of the manuscript)

This has been addressed.

L206: Typo "underestimate the show the correct..."

This has been addressed.

**Responses to reviewers following submission of manuscript
“The relative role of direct orbital forcing, and CO₂ and ice sheet
feedbacks on Quaternary climate” by Williams *et al.***

REVIEWER COMMENTS

We extend our sincere appreciation to all Reviewers for their examination of our manuscript. Here, we address their suggestions, comment-by-comment. In the below, reviewers' comments are blue and in a smaller font, and our corresponding response follows in a standard font. Line numbers in the following reviewer comments refer to the previously-submitted version of the manuscript, whereas line numbers in our responses refer to the new version of the manuscript. We have also have numbered the comments below and provided references to these as comments next to the text (main manuscript and supplementary material) to aid readability.

REVIEWER COMMENTS

Reviewer #1 (Remarks to the Author):

Williams et al 2024 Nature
Second phase review

The manuscript is much improved from what was submitted previously.

0. We thank the reviewer for their comments.

However, I reiterate here as I have to the editor that I am an emulator adjacent scientist. This manuscript absolutely must be evaluated by a researcher who actively uses emulators and is familiar with the quickly changing field of AI/ML in climate science. Although I work with colleagues who are experts in emulators and feel very familiar with their use and applications, I do not use them in my own research and thus do not feel qualified to comment on the particulars of the technique applied here.

0. A third reviewer was appointed, and their comments addressed (see below). The second reviewer was satisfied with our previous responses (see below).

As in my previous comments, there are many important feedbacks within the earth's climate in modulating the climate response. The PMIP/CMIP literature acknowledges this fact with its 'spur' experiments: lgm lgm+dust midholocene midholocene+veg (plus methane lakes etc) ... ismip experiment ... volmip experiment ... nahosmip ... In the first paragraph of the intro, the things your study can address are listed, but the things you did not consider are not discussed. Be clear to set up the limitations and expectations here and realistic about what the implication of these exclusions. These caveats belong up top, in the first paragraph or two, and you can even set up a nice resolution of the outstanding question- whether the exclusion meant you missed out on important features in the transient. It is especially disappointing to have 3 of the forcings being the orbital parameters—which are then determined to be unimportant using the M score (which isn't quite believable). And it isn't clear to me that using the parameter coefficients of orbital forcing versus some derived quantity for insolation is correct. More details on orbital forcing are required for such a ringer conclusion (contrary to pretty much all the paleoclimate literature out there) about orbital forcing not being the driver and most important forcing. And as a set, since CO₂ is how orbital forcing gets parsed through the carbon cycle and ice sheets are how that gets parsed into the cryosphere, perhaps it IS better to use other derived quantities like vegetation or dust or 1st derivative of sea level (ice melt) or similar as the forcings instead of the raw orbital parameters.

1. Firstly, a sentence has been added towards the end of the introduction (lines 135-141), detailing which feedbacks/processes are included in the HadCM3 model (and hence included in the emulator), and which are not included.

2. Secondly, we have clarified what we mean when we say that the CO₂ and ice feedbacks are the dominant drivers (rather than the orbital forcings), both in the Abstract (lines 41-45) and the Discussion (lines 403-418). To clarify, we are not saying that the net overall effect of three orbital forcings are unimportant – indeed, because CO₂, ice and other feedbacks are ultimately driven by these orbital forcings, the net overall orbital forcing is, of course, very important. However here we are looking at the direct effect of the orbital forcing, which, despite being very large seasonally, is relatively small on the annual mean, resulting in a relatively small impact on annual mean temperatures.

The methods section is much better now. But the main text readability isn't that great. Too much in the weeds about certain things without enough detail to follow the argument for others (details below). Stylistically, I think the main text needs a strong editing hand.

We have made a general edit of all the main text, including throughout the Introduction, Results, Discussion and Conclusions.

I also did ask that the authors make plenty of comparisons to the Yun et al 2023 simulation which is the most similar to their efforts. They have done that—but that is not to say that this should be to the exclusion of other comparisons and efforts.

3. We have added in more studies undertaking similar comparisons and efforts, in the Introduction (lines 67-71 and 93-95).

There is an implicit question through the text that needs explicit address:: hysteresis. That is to say, does the climate response depend on the initial state. This question comes up in the discussion about the relative merits of the speeding-up technique used in the Yun 2023 manuscript versus the emulator. Does the emulator include a phase delayed temperature (CO₂ et al) forcing field or anything else to include initial state? Or is it judging each climate state independently.

4. Here, we judge each climate state independently, therefore the issue of hysteresis is not considered. We do, however, acknowledge that this is an interesting aspect, and could be explored in future work. A sentence has been added in the Discussion to highlight this limitation (lines 435-439).

It STILL appears that there are two separate emulators (glacial and interglacial) that are stitched together. Is it any wonder that the main finding of ice sheet is most important is arrived upon when its baked in to the process? I remain concerned, even after reading the responses and rewritten methods section, that there are two distinct emulators. Without this continuum, what chance is there to capture changes at the mid-pleistocene transition? Authors aren't using a linear regression that might suffer from assigning warming of similar magnitude to cooling from G to IG states. That's the point of this method. I am not convinced about the use of two separate emulators. What happens if they use just one. L529 'steps were taken' -> this is the methods section to describe those steps.

5. As we explain in the Methods (lines 576-593), we were concerned about using a single emulator due to possible unphysical 'leakage' effects from the glacial to the interglacial, resulting in unphysical warming in the region of the Laurentide, Fennoscandian, and Antarctic ice sheets during interglacials. In order to test this, we created a combined emulator, i.e. the glacial and interglacial conditions were considered as a single emulator. This combined emulator is illustrated in the following figure (now also included in the Supplementary Material, Figure S1e), which shows each of the HadCM3 training simulations according to the relative sea level, log CO₂ and insolation at 65°N (combining the three orbital parameters into a single parameter, just for the purposes of the figure). Each training simulation, represented by the dots, has been categorised according to whether relative sea level is at (red), above (yellow), or below (blue) 0 m.

We repeated the ‘all drivers’ simulation (i.e. where all five drivers varied according to proxy data) but using this combined emulator. This can be seen in the figure below.

When compared to Figure 1 in the manuscript, there is little difference at the SST locations (panel b-e) but substantially more uncertainty in SAT over Antarctica (panel a) when a combined emulator is used.

We therefore consider it appropriate to use the two separate emulators, as combining them into a single emulator does indeed lead to higher uncertainties at the boundary between

glacials and interglacials, likely due to the ‘leakage’ effect discussed in the text. All of this is now discussed in the Methods (lines 576-593).

I still remain concerned about emulating AMOC as per the first review. I do not see this addressed in the main text, not relegated to a mention deep-deep in the methods L542. AMOC remains one of the confounding factors for folks working on climate model emulators and it is odd to not explicitly address this piece.

6. The emulator is a purely statistical model designed to replicate the output of HadCM3, and as such does not *explicitly* include any representation of any physical processes, including the AMOC (or indeed any other process, e.g. atmospheric circulation, clouds, sea ice etc.). However, because physical processes are represented in HadCM3, the output of the emulator does implicitly account for these processes, including the AMOC. This has been made more clear in the Introduction (lines 141-145).

Major revisions are still needed, although somewhat less major than last round. This emulator exercise is important for the paleoclimate community, although I am not convinced the present work is complete enough as presently written (and possibly done for the forcing comments above).

7. Revisions have now been made throughout the main text, both according to comments here (and Reviewer 3) and comprising more general edits.

L86: This is where you should explicitly state that this type of simulation assumes that the model simulation is relatively insensitive to initial conditions – you need to also discuss your own initial conditions. Was the training HadCM3 mindful of initial conditions? (Did you start your Pliocene simulations from an existing Miocene simulation or was it initialized from a default configuration.) If it was not, then you also have this same potential <lack of> initial condition bias built in.

8. All of the HadCM3 training simulations were initialised from preindustrial conditions and then spun-up to quasi-equilibrium; this means that the training data can be considered independent of the initial conditions of the underlying GCM simulations. This is now discussed in the Methods (lines 556-560). The glacial-interglacial timeseries presented in the paper are also independent of the initial emulated simulation in the timeseries, because each emulated time period within the timeseries is an independent snapshot, stitched together to make a timeseries.

Hysteresis (aka tip and the like) is a major point of concern for future comparisons. iow: I would frame this comparison a bit differently. It does nothing to degrade your work by having another group achieve a similar result—rather it gives an excellent basis for comparison with strengths and weaknesses of your (their) approach, (in)consistencies, etc.

9. This has already been addressed by the above comment (point 4) and in the Discussion (lines 435-439), where we discuss in the main text that hysteresis is not considered here.

L103-5: “simulate” => “interpolate” ◊ this is what you are doing, you are not simulating. Emulators interpolate points determined by data or physics-based models, and totally dependent on the training data “points”. The word “emulator” denotes the relatively new application of new AI/ML techniques instead of traditional to the statistical models reproduction—please introduce this concept in this paragraph.

10. A sentence has been added in the Introduction to clarify the above point (lines 116-118). We have also been through the manuscript and ensured that when we are talking about the results from the emulator, we are clear on this e.g. by using ‘emulated’ instead of ‘simulated’.

L119: “proxy” => “past climate” using “proxy” as a noun becomes ambiguous as you cross fields.

11. This has been removed as part of other edits (line 129).

L120: “physics-based”, more complex GCM...

12. This has been added (line 113).

L121: the number of simulations used in the training data is related to the number of parameters explored – use the specific language. What did you use to properly sample the phase-space? Latin Hypercube? Something else? Introduce here. How many simulations are required per parameter?

13. This paragraph has now been moved into the Methods (lines 486-506), in line with Reviewer 3’s comments that the main text should focus more on the results. But we have also added to this sentence, to clarify that Latin hypercube sampling was indeed used and is discussed in more detail below (line 489-490).

L126-128: In reading this section, the word parameter feels muddled with the word parameter as it is used in perturbed physics ensembles—those take on the PPE acronym. Perhaps there is a use-specific acronym? Boundary-Condition-Parameter (BCP)? This technique has been applied to sampling the parameter space used within the model physics (eg doi:10.22541/essoar.172745119.96698579/v1), to interpolate between future earth scenarios (eg <https://arxiv.org/html/2404.13227v2>)

14. This sentence has now been moved into the Methods (lines 492, 496), in line with Reviewer 3’s comments that the main text should focus more on the results. But we have also removed the word ‘parameter’ from this line, to avoid confusion (line 496). We have also been through the manuscript and ensured we are consistent with our terminology, such as the use of ‘driver’ versus ‘forcing’.

L141: This stat of ‘5 minutes to run XX years’ will not age well and there is not enough info about the emulator in-line to know how complex it is (Also- is the emulator full atmosphere 3d, ocean 3d?, land surface 3d?). Framing in comparison to the amount of time required to run the full model is sufficient.

Further, the emulator is solving each state as if it is totally independent (my take after reading the methods but this should be spelled out briefly in the text) — which is equivalent to doing time slices. Authors use 1kyr resolution, which is equivalent to doing 2580 time slice simulations. The emulator was trained on 182, 500 year time slices. Since they provide 120 years / day as the length scale of these simulations, these training simulations took <5 days. If the runs were done not overlapping, that is 29 years. Running the ensemble at a more typical few dozen at a time, that would be a run time on order of a year or so. In any case, there is too much in the weeds on this run time justification in the main text and not enough details on the emulator in the main text.

15. This paragraph has now also been moved into the Methods (lines 508-524), in line with Reviewer 3’s comments that the main text should focus more on the results. But we have also rephrased this sentence and taken out any reference to an exact number of minutes (lines 508-524).

L151: Does HadCM3 really output subdaily diagnostics and then average those into monthly averages? That is an interesting choice. I would say most groups do not do that I/O step because I/O tends to slowdown the simulation A LOT because you are stopping and writing to the disk and depending on the filesystem that can be rather slow. Many groups keep up with tracers in memory at the normal timestep (20-30 min), and then only output at the temporal resolution required, vastly decreasing I/O. Are the authors positive that HadCM3 does not work that way? The next few sentences are too much in the weeds for main text and should be moved to the methods or supplemental.

16. This sentence been removed, and the whole paragraph has been moved into the Methods (lines 508-524). HadCM3, as with all GCMs, makes calculations and updates the system state every model timestep (15 minutes in the atmosphere). However (also like all GCMs), it

does not output all these 15-minute states to disk. Instead it stores these in memory and then calculates the monthly-mean at the end of the month, and outputs this to disk.

L189: The discussion of figure 1 is too qualitative. Did you plot and calculate $T_{emulator}$ versus T_{proxy} ? The figure 1 plot here takes up too much space and yet the curves are really too tiny and delicate to ‘tell’ what the authors are talking about.

17a. Firstly, we have increased the size of Figure 1, and we now feel the curves are easier to see. Secondly, we have added a sentence (line 184) in the Results pointing the reader to the quantitative discussion of this figure, further down in the results when the M scores are introduced. Finally, we have added a figure into the Supplementary Material (Figure S0), showing a scatterplot of emulated temperature versus proxy temperature for the same five locations as Figure 1, and discussed this in the Results (lines 187-190).

There also needs to be a context of traditional ‘model’ built in. If you simply regress your forcing fields on (say) the 3d SST field, how does this statistic compare with the emulator’s? $T_{regress}$ versus T_{proxy} ? For the d18O_{ice} record, how do your results stack up against the EPICA Grand Challenge results (where simple and fast box models were applied to link CO₂ and temperature for Antarctic ice core).

17b. This sounds interesting, but correlating just one of the forcings (e.g. CO₂) with the site-specific proxy temperature is too generous to the emulator, because the emulator ‘sees’ all 5 forcings (CO₂, sea-level and the 3 orbital parameters). Doing something more complex, in which for example we build a multi(5)-variate regression between the forcings and the site-specific temperatures (as was done in some of the responses to the EPICA Challenge; Wolff *et al.* 2011), is outside the scope of this paper and, furthermore, such a regression-based model is contrary to the ‘philosophy’ of our study, in which the underlying GCM is entirely mechanistic, and proxy temperature observations play no part in the model development or tuning; the model-data comparison in our study is a purely emergent property.

With these other methods, do you have the same bias periods that you lay out on L193?

17c. The scatterplot now shown in the Supplementary material (see above comment) also identifies the three MISs discussed in the Results, where the same biases are evident i.e. during these periods, the proxy data are generally warmer than the emulated temperatures. This is now discussed in the Results (lines 193-195).

On L211: 4degC is FAR too little cooling for last glacial maximums. This magnitude should be set by data assimilation methods eg <https://doi.org/10.1038/s41586-021-03984-4> and in context of PMIP3/PMIP4 LGM models.

18. This comparison been removed, as it is merely demonstrating the observation above (line 209). Global means from the various LGM datasets are discussed in the Supplementary Material (lines 86-90) and are as follows (all values show LGM cooling relative to the PI): emulator (this paper) = 5.1°C, HadCM3 (this paper) = 5.3°C, Annan and Hargreaves (2013) = 4°C and Osman *et al.* (2021) = 6-7°C.

L 204/701: You talk about the interpolated CO₂ up to 800k in this study and then punt to the methods. Probably better to put concisely enough in the text to follow what you’ve done (we interpolated 10k resolution to 1k resolution etc). You use an outdated source for CO₂ prior to 800k. I recommend <https://doi.org/10.1126/science.adi5177> which is the recent Dec2023 CenCO₂PIP consensus paper.

19. A sentence has been added in the Methods (lines 711-717) to include the above reference, to justify why we did not use it and instead used our existing CO₂ source (Van de Wal *et al.* 2011), and to clarify that we have interpolated the data to 1 kyr.

L217: Are you saying that you have the same as the Yun paper or the same variability. It isn't clear.

20. This has been corrected (line 216).

L221: Again here, this would show up better in a scatter plot to augment figure 1 as the highest and lowest values having the greatest error.

21. This has been corrected by the comment above.

L232: Can you say the interpretation something like "G-IG CO₂ would need to be 1.5 times larger to match the proxy G-IG temperatures (Figure S6)" instead of this hand waving that requires digging into the supplemental. After looking at Figure S6, I am not entirely sure what you've done. Do you not have actual CO₂ in 800k to 1.1k? Are you comparing optimized CO₂ to actual CO₂? Can you put numbers on this?

22. This sentence has now been modified to incorporate the above (lines 231-234).

L239: OR your mismatch could be that you have not included enough forcing fields. Like vegetation or etc written above.

23. This sentence has now been modified to incorporate the above (lines 244-245).

L243: MAP is defined above, but it is an uncommon shorthand and makes the paper hard to read. SAT, in contrast, is a common acronym. Why not use PREC to improve readability? Also, Chinese speleothem d18O are not MAP. They are changes in d18O that memorializes monsoon strength.

24. This has now been done (line 247 and throughout the rest of the manuscript), and a sentence has been added to clarify that Chinese speleothem d18O are not mean annual precipitation but that we are using it as a first-order comparison here (line 247-249).

L262: 'all drivers' simulation (E11111) Each subscript means forcing on or off? Which is which? Is this spelled out somewhere? Although this shorthand likely works great in the lab, it is hard to read.

25. This has now been clarified when this simulation is first mentioned at the beginning of the Results (lines 162-163).

L265: as above, some sort of basic regression model comparison/context should be added.

26. This has been corrected by the comment above.

L269: I don't follow what this means. Constant climate signal of what? Do you mean "equilibrium" climate experiment like PMIP? If that's what you mean, what were their scores?

27. This has now been clarified, directly in the Results (lines 275-276) and also linking to the relevant part of the Methods where this is discussed (lines 757-763).

L289: You cannot punt folks to the supplemental text and then descend into unfamiliar jargon like "where each driving component is averaged over all 120 pathways" I have no idea what you are talking about. The supplemental should AUGMENT the main text, not render it unreadable. When you read in the supplemental the full suite of runs—decode them here otherwise readers have to go back and dig into the methods (as per comment above about E11111), and then dig back into the original point in the paper. This isn't readable. Please add a supplemental table what E10000, E01000... E00001 mean.

28. This sentence has now been removed from the Results (lines 295-300) and a table has been added to the Supplementary Material (Table S3) and pointed to in the Methods (line 803).

L293: If you get a result that the impact of the orbital forcing on M scores is negligible—perhaps this is an indicator that the orbital ‘forcing’ the way you’ve implemented it isn’t quite right. It is the impact that these have on insolation seasonality that is important. And it seems weird to have eccentricity separate from precession as two separate forcings.

29. We have added a sentence in the Results (lines 304-307), firstly to clarify that eccentricity and precession are not two separate forcings, and secondly to reiterate (as already discussed in the Discussion, see above comment) that we are not suggesting that the overall net orbital forcing is negligible, but that the direct orbital radiative forcing on the annual mean temperature response is very small. This result is in line with previous work (see Discussion, lines 403-406).

What is the M score of the orbital parameters to CO₂ and ice sheet volume? These need to be plugged in to a function to yield insolation. Does the M score technique capture that?

30. The M score measures the goodness-of-fit of emulated temperatures (in °C) relative to observations of temperature (in °C). It is not possible to construct an M score of two forcings (with different units), relative to each other. We have added a sentence to clarify this in the Methods (Metric for quantifying model performance, lines 813-814).

Perhaps you should use a variation of the original specmap methods (insolation at 65N) Perhaps max-min insolation at mid latitudes of northern and southern hemisphere (or something to indicate seasonality magnitude) plus a baseline total insolation at (say) 65N?

Given that the emulator is so inexpensive to run, it would seem that you could experiment with other forcing parameters.

Or perhaps this is an indication that the orbital forcing is already built in to the CO₂ and ice volume signal so well that they are redundant? And maybe these 3 parameters should be swapped out with vegetation. Vegetation could be distilled down to the midHolocene pmip variations (fraction of ‘green sahara’ and shrubs at high latitudes). Or maybe other things like aerosol loading might be used as a forcing (as CO₂, easier done 800k to present than before). Or perhaps use the first derivative of the sea level (something like amount of ice melt in NH and SH).

31. Addressing all 3 of the above comments together, the orbital forcing is calculated from Laskar *et al.* (2004), implemented into the full HadCM3 training-data simulations. This includes the complete seasonal and latitudinal evolution of insolation, giving us the direct orbital forcing. As the reviewer states, and as we now state in the Results (lines 309-313), the indirect impact of orbital forcing is implicitly included in the CO₂ and ice sheet forcings.

In short, do not think this paragraph L297 with the orbital parameters causing <5% of the signal is defensible as written and goes against 35 years of prevailing thought.

32. In our view, this statement does not go against 35 years of thought. It is very well established, and consistent with our results, that direct orbital radiative forcing leads to feedbacks associated with CO₂ and ice sheets, which in turn result in glacial-interglacial cycling. However, the direct radiative impact of orbital forcing itself (which is very close to zero on an annual-global mean), and in the absence of CO₂ and/or ice sheet feedbacks, has relatively little effect on temperature (in particular, on annual-mean temperature). This has previously been shown in simulations of specific time periods, such as during the LGM (e.g. Cao *et al.* 2019, Shi *et al.* 2023), but here we show it for the entire 2.58 Myr. As per the reviewer’s first comment, this is now discussed in the Discussion (lines 401-418).

L346: Why are there only modern and lgm ice? I know that there are HadCM3 deglacial experiments. I realize that I made this comment before, and there is a response to reviewer, and yet I am not satisfied given that the

lack of ice variability is one of the issues discussed in the discussion.

33. There is not only LGM and modern ice in the HadCM3 training simulations. In addition to modern ice (the 60-member *modice* ensemble), the training data includes the glacial-interglacial cycle simulations of Singarayer and Valdes (2010) (the 122-member “*highice*” ensemble), and also Pliocene-like simulations (the 60-member “*lowice*” ensemble). A sentence to clarify this has been added in the Methods (Creating the interglacial and glacial training data, lines 571-574) – see also the figure of the combined emulator above which shows this clearly.

L642: You should use BOTH the updated Hargreaves/Annan estimates (Renoult) generally on the low-end of accepted across the community and the Osman/Tierney estimates generally on the higher end of accepted across the community from data assimilation methods here instead of this older work.

34. This figure has now been modified, so that it includes the more recent Osman *et al.* (2021) data assimilation method, in addition to the other 3 panels (emulator, HadCM3 and Annan and Hargreaves (2013)). We have removed this discussion from the Methods (lines 663-669) into the Supplementary Material (specifically Section S4 (SM, lines 68-96), where we discuss this addition alongside the other LGM panels.

Reviewer #1 (Remarks on code availability):

No I didn't look through the code this time.

35. We believe we have satisfied all of the comments concerning code, from both the reviewer and the editor.

Reviewer #2 (Remarks to the Author):

No more comments (all comments have been answered in the new version of the paper)

1. We thank the reviewer for their previous round of comments.

Reviewer #3 (Remarks to the Author):

The idea of a paleoclimate emulator is a good one. These simulations are so long and expensive that we already (often) use pared down climate models, and this is taking it another step to pare things down even further, which is a worthy goal. The emulator seems to be well constructed and carefully validated. In general, I really like this manuscript. However, I have a question about whether it fits the scope of the journal.

2. We thank the reviewer for their comments.

Where I struggle with this paper is that it seems to be focused on describing and validating the emulator's performance, which is what I might expect for a more methods-focused journal. What I think is lacking are insights that the emulator can provide. Examples could include figuring out what processes are important to represent in models for understanding paleoclimate variability or revealing drivers of past changes. The authors do point out that CO₂ concentration and ice are more important than the inputs corresponding to orbital parameters, but given the way the results are presented and caveated, I can't tell whether those results are new or surprising, or even whether they are simply a product of what was done.

3. We have modified a number of areas to address this comment. Firstly, we have rewritten the abstract (lines 33-45) so that it focuses more on the results, and less on the methodology. Secondly we have likewise modified the Introduction (lines 64-71 and 129-155), and the Conclusions (lines 446-447 and 469). We have also removed some of the methodology previously in the Introduction, summarising this here (lines 130-155) and moving the full

description to the Methods (lines 486-506). Finally, we have moved the description of the emulator's efficiency out of the Results and into the Methods (lines 508-524).

My hope is that these issues can be taken care of via wording and do not require much new analysis. The paper is quite well done, and the tool the authors have produced is quite valuable.

We hope that all of the above changes mean that the main part of the manuscript now focuses less on the methodology and emulator validation, and more on the primary research question of the driving mechanisms of Quaternary climate change.

The authors also did an excellent job of responding to the previous review comments, some of which were quite challenging.

We thank the reviewer for their comments.

Reviewer #3 (Remarks on code availability):

The other reviewers focused on the code review, so I didn't do this part.

We believe we have satisfied all of the comments concerning code, from both the previous reviewers and the editor.

**Responses to reviewers following submission of manuscript
“The relative role of direct orbital forcing, and CO₂ and ice sheet feedbacks
on Quaternary climate” by Williams *et al.***

REVIEWER COMMENTS

We extend our sincere appreciation to both Reviewers for their examination of our manuscript. Here, we address their suggestions, comment-by-comment. In the below, reviewers' comments are blue and in a smaller font, and our corresponding response follows in a standard font. Line numbers in the following reviewer comments refer to the previously-submitted version of the manuscript, whereas line numbers in our responses refer to the new version of the manuscript.

REVIEWER COMMENTS

Reviewer #1 (Remarks to the Author):

Review of Williams et al 2025

This paper is greatly improved from previous versions. I do not think I will need to see this again before the editor makes a decision.

We thank the reviewer for their comments.

1) I'm still not satisfied with two stitched together emulators being used instead of one. It makes the very interesting and good work seem less so because the answer gets baked in to the training. *“The approach of having two separate emulators was taken to ensure that there was no ‘leakage’ of state-specific climate variations across different climate states (see Supplementary Material, Figure S1e). For example, under glacial conditions development of the Laurentide ice sheet is accompanied by strong cooling over northern North America. However, this does not mean that during interglacial conditions there should be a strong warming in the same region, since the data are presented as an anomaly from interglacial present-day conditions (where no ice sheet is present).”*

Actually, the change in height alone does mean there is rightfully a huge amount of warming in this area. In anomaly space this does look odd—it just does. If you want to get rid of that, try using vertical lapse rate to apply a correction.

In the previous iteration of the manuscript we qualitatively explained this leakage effect in terms of the impact of the Laurentide ice sheet (lines 576-593 of the previous iteration of the manuscript), but the effect is even clearer over Greenland and Antarctica. We now quantify and illustrate (new Figure S10) this leakage effect in the revised Methods section (‘Step 2’, lines 585-615), showing clearly and definitively why two separate emulators are required, as follows:

“The approach of having two separate emulators (shown in the Supplementary Material, Fig. S9) with separate training data for the ‘interglacial’ emulator applied for $GSL \geq 0$ m compared with the ‘glacial’ emulator for $GSL < 0$ m, was taken to ensure that there was no ‘leakage’ of state-specific climate variations across the glacial-interglacial transition. The concept and effect of this leakage can be illustrated (see Figure S10 in the Supplementary Material) by considering surface air temperature for two climate states, one with an emulated climate with $GSL = +5$ m (Fig. S10a and b), and one with an emulated climate with sea level = -5 m (Fig. S10c and d), firstly as produced by the separate emulators (Fig. S10a and c, as used in this paper) and secondly produced by a combined emulator, in which all

training data are used to train a single emulator (Fig. S10b and d). For the +5 m climate state, both the separate and combined emulators show clear warming associated with melt of the Greenland and Antarctic ice sheets (Fig. S10a and b). This is correct, and appears because the “lowice” training data include ice sheet reduction in these regions, associated with the PRISM4 mid-Pliocene ice sheet reconstruction. However, for the -5 m climate state, there are clear differences between the results of the separate and combined emulator versions in these regions (Fig. S10c and d). For the combined emulator (Fig. S10d), the algorithm is still influenced by the Pliocene “lowice” training data, producing anomalous cooling in the regions that show melt in the “lowice” training data. However the two separate emulators are, for this GSL, only influenced by the “highice” and “modice” training data, and so do not have this unphysical “leakage” across the glacial-interglacial transition (Fig. S10c). In essence, the use of separate emulators is required because in the “real” climate system, warming relative to modern results in ice sheet loss primarily in the Greenland/Antarctica region, whereas cooling relative to modern results in ice sheet growth primarily in the Laurentide/Fennoscandian region. The use of two separate emulators therefore ensures that the different states are characterized separately and correctly. In addition, checks were carried out to ensure that there were no discontinuities across the glacial-interglacial transition between the climate projections from the two different emulators. This lack of discontinuity is unsurprising, given that the “modice” ensemble was common to both emulators, and sits at the surface $GSL=0$, on the boundary of the two emulators. We therefore consider it appropriate to use the two separate emulators, as combining them into one leads to higher uncertainties at the boundary between glacial and interglacials.”

Fig. S10. Surface air temperature anomalies (compared to the preindustrial control (i.e. 0 kyr), °C) for two climate states, and using two approaches to running the emulator: a) climate state with $GSL = +5$ m, separate

emulator approach; b) climate state with GSL = +5 m, combined emulator approach; c) climate state with GSL = -5 m, separate emulator approach; d) climate state with GSL = -5 m, combined emulator approach.

2) If the authors used a single emulator – show it. It makes no sense the explanation that there is leakage over North America and then differences between the two emerge on the opposite side of the world “*When compared to proxy data at the five sites presented in Figure 1, the results showed no difference in SST but higher variability for SAT over Antarctica (not shown).*”. Show both emulators.

Given the above explanation, we have removed the sentence referring to this additional figure (lines 612-615) because, as discussed, using a combined emulator produces incorrect results.

3) The “M score” is not described adequately for readability in the text. In the main text, just a few short words like “M-score (Arcsin-Mielke measure M) quantifies the goodness of fit of the emulator to the proxy climate archives such that where no skill is 0 and maximum skill is 1000 (see methods)” will allow the reader to keep reading without “flipping” to the back page.

We have added some text into the Results section of the main manuscript (lines 264-269) similar to the above suggestion, and removed similar text from the methodology to avoid repetition (lines 835-843).

4) The authors insist on using the 2013 Annan and Hargreaves extremely low LGM number which no one believed at the time, and not even they believe today, as memorialized here
Annan, J. D., Hargreaves, J. C., and Mauritsen, T.: A new global surface temperature reconstruction for the Last Glacial Maximum, *Clim. Past*, 18, 1883–1896, <https://doi.org/10.5194/cp-18-1883-2022>, 2022.
“*Abstract: We present a new reconstruction of surface air temperature and sea surface temperature for the Last Glacial Maximum. The method blends model fields and sparse proxy-based point estimates through a data assimilation approach. Our reconstruction updates that of Annan and Hargreaves (2013), using the full range of general circulation model (GCM) simulations which contributed to three generations of the PMIP database, three major compilations of gridded sea surface temperature (SST) and surface air temperature (SAT) estimates from proxy data, and an improved methodology based on an ensemble Kalman filter. Our reconstruction has a global annual mean surface air temperature anomaly of °C relative to the pre-industrial climate. This is slightly colder than the previous estimate of Annan and Hargreaves (2013), with an upwards revision on the uncertainty due to different methodological assumptions. It is, however, substantially less cold than the recent reconstruction of Tierney et al. (2020). We show that the main reason for this discrepancy is in the choice of prior. We recommend the use of the multi-model ensemble of opportunity as potentially offering a credible prior, but it is important that the range of models included in the PMIP ensembles represent the main sources of uncertainty as realistically and comprehensively as practicable if they are to be used in this way.*”

Just take Annan and Hargreaves 2013 out. Replace it with something folks DO believe. (And many people still believe that -4.5degC is on the low end.)

This has been done, removing Annan and Hargreaves (2013) and replacing with the data assimilation reconstruction of Tierney *et al.* (2020), for Figure S3 and in the text where this is discussed (Supplementary Material, Section S1, lines 11-33).

5) The LGM reconstruction of Ice5G is not a very good one. When it came out, no one believed it either – particularly people studying the Laurentide Ice Sheet. They did not believe it because it pulled all of the downsizing ice from the Fenno-Scandian and Antarctic and piled it high on the Laurentide. Unbelievably so. Ice 4G and 6G are perhaps defensible, but 5G is not. The climate regimes by such a large ice mountain are just different from those without it.

We appreciate that there is some uncertainty in the configuration of the ice sheets, including for the LGM. However, it is outside the scope of the manuscript to fully explore this uncertainty, as this would require hundreds of additional GCM simulations. However, we do discuss this uncertainty in the Methods section of the main manuscript (‘Step 2’, at lines 617-626) and the Supplementary Material (Section S3, lines 73-84), as follows.

Main manuscript: “The reconstructed ice sheet extents in the “highice” ensemble are based on the ICE-5G model of Peltier (2004). These reconstructions were primarily selected because the associated HadCM3 model simulations already existed (Singarayer and Valdes 2010). The reconstructions are based on palaeo data (GSL, local sea level, and ice-sheet extent) and include the range of associated data that was required for this work. In addition, for glaciations prior to the last glacial cycle, there is very little or no palaeo data available that enables the global 3-D reconstructions which are required here. There are, however, other reconstructions (see Section S3 in the Supplementary Material); although we acknowledge the existence of these, the “lowice”, “modice” and “highice” ensembles used here, when combined, do capture changes in climate and ice sheet extent ranging from interglacial states to full glacial conditions.”

Supplementary Material: “Other ice sheet extents reconstructions are available (see, for example, Schmidt et al. 2014), as well as results from recent ice sheet models (see, for example, Tarasov et al. 2025), which could result in changes to the simulated climate. For example, some studies have suggested a lower maximum elevation for the Laurentide ice sheet during glacial conditions (e.g. Abe-Ouchi et al. 2015). Modelling studies have suggested that the topography of the ice sheet can have a significant impact on mean sea level pressure, which affects the location of the wind-driven gyre circulation in the subpolar North Atlantic and on the strength of the AMOC, resulting in warming in the North Atlantic (Colleoni et al. 2016). Storm tracks in the North Atlantic have also been shown to be affected, with associated impacts on precipitation and snowfall in northern Europe (e.g. Colleoni et al. 2016). However, we do not expect these uncertainties to impact our main conclusions in terms of the relative importance of the various drivers.”

6) The paper reads much better. And I'm glad the authors took the suggestion to cite more works beyond Yun 2023. But now the discussion slogs through Erb 2015 and it starts to read like a review instead of a nature article. There are many more models and simulations beyond Erb and Yun that have explored the ice sheet height and sensitivity of the climate.

We have made the discussion of Erb et al. (2015) more concise while keeping all the salient points, and have included some other related modelling studies (lines 383-411).

7) I thank the authors for their clear explanation “CO₂, obliquity, eccentricity, precession and ice, where 0 = constant PI values and 1 = varying values. Hence E00000 (the ‘no drivers’ simulation) has all drivers set to constant PI values, E10000 has varying CO₂ but everything else is set to constant PI values, and so on, up to E11111 where all forcings are varying (the ‘all drivers’ simulation).” In addition to the (correct) explanation that eccentricity and precession are convolved, why have 5 subscripts instead of 3 (let the last three be ‘orbital’)? The authors state plainly that eccentricity is not tested without precession. Is obliquity tested separately? If obliquity IS tested separately, it would be very interesting to know if obliquity is more important than precession/eccentricity as obliquity has been proposed as the cause of glacial cycles (Huybers etc). It would also be interesting to know if this was not the result.

We need 5 subscripts because in the sensitivity studies we are individually testing the sensitivity to the 3 orbital components, plus ice sheets, plus CO₂; therefore, 5 subscripts are needed to fully describe the relative contribution of each orbital individual driver and the ice and CO₂ drivers, which necessitates the ability to ‘turn on or off’ any or all of the 5 drivers independently. We have clarified this in the Results section of the main manuscript (lines 307-309). The obliquity is therefore tested separately, and the results of this are shown in Figure 3; here, compared to eccentricity and precession, obliquity is indeed providing a higher contribution, particularly over high latitudes e.g. the Southern Ocean. This is discussed in the main manuscript (lines 305-307). However, we do not discuss this in the context of Huybers & Wunsch (2005) because here we are just looking at direct orbital

forcing, rather than the impact of orbital forcing on CO₂ and ice sheets, which is important for the Huybers hypothesis.

8) The presentation of figure 1 is improved—but I still find the lines too delicate – are the barely visible fuzzy nature the uncertainty? It is tricky to have a multipanel plot with the much larger Antarctic temperature shifts and the b-e panels then scaled way too zoomed out. There is still much white space and what feels like default subplots. This can still be done better – literally most paleoclimate folks do this with having no extra x-axes, moving the curvy lines physically closer together, alternating y1 and y2 axes labels, and expanding the smaller magnitude to be visible (just make sure to note clearly that this is what you've done).

This has been done: removing the whitespace, changing the y-axis scales to make each appropriate for each location, alternating the y-axes and removing the multiple x-axes.

9) Figure 3. Figure S2. Figure S4. Figure S8. Use a projection like Robinson (etc) that doesn't blow up the poles so much. I really have no idea why all the scripting languages let awful projections be the default.

This has been done, with all of the above figures reproduced using the Robinson projection.

Reviewer #3 (Remarks to the Author):

The authors have done a nice job, and I have no additional comments. Reviewer #1 did raise some important questions that seem to be satisfactorily answered, but I'll happily defer to them.

We thank the reviewer for their previous round of comments.